# Breakups are Complicated: An Efficient Representation of Collisional Breakup in the Superdroplet Method

Emily de Jong[1], John Ben Mackay[2,*], Oleksii Bulenok[3], Anna Jaruga[4], and Sylwester Arabas[5,**]

[1]Department of Mechanical and Civil Engineering, California Institute of Technology, Pasadena, CA, USA
[2]Scripps Institution of Oceanography, San Diego, CA, USA
[3]Faculty of Mathematics and Computer Science, Jagiellonian University, Kraków, Poland
[4]Department of Environmental Science and Engineering, California Institute of Technology, Pasadena, CA, USA
[5]Faculty of Physics and Applied Computer Science, AGH University of Krakow, Kraków, Poland
[*]Research carried out in part while at the California Institute of Technology (Pasadena, CA, USA)
[**]Research carried out in part while at the University of Illinois (Urbana-Champaign, IL, USA) and at the Jagiellonian University (Kraków, Poland)

**Correspondence:** Emily de Jong (edejong@caltech.edu)

**Abstract.** A key constraint of particle-based methods for modeling cloud microphysics is the conservation of total particle number, which is required for computational tractability. The process of collisional breakup poses a particular challenge to this framework, as breakup events often produce many droplet fragments of varying sizes, which would require creating new particles in the system. This work introduces a representation of collisional breakup in the so-called "superdroplet" method which conserves the total number of superdroplets in the system. This representation extends an existing stochastic collisional-coalescence scheme and samples from a fragment-size distribution in an additional Monte Carlo step. This method is demonstrated in a set of idealized box model and single-column warm-rain simulations. We further discuss the effects of the breakup dynamic and fragment-size distribution on the particle size distribution, hydrometeor population, and microphysical process rates. Box model experiments serve to characterize the impacts of properties such as coalescence effieciency and fragmentation function on the relative roles of collisional breakup and coalescence. The results demonstrate that this representation of collisional breakup can produce a stationary particle-size distribution, in which breakup and coalescence rates are approximately equal, and that it recovers expected behavior such as a reduction in precipitate-sized particles in the column model. The breakup algorithm presented here contributes to an open-source pythonic implementation of the superdroplet method, 'PySDM', which will facilitate future research using particle-based microphysics.

## 1 Introduction

The superdroplet method (SDM) for cloud microphysics is a high-fidelity particle-based (Lagrangian) representation of aerosols and hydrometeors that offers notable advantages over traditional bulk and bin microphysics schemes. Particle-based methods were initially used in atmospheric simulations to represent ice nucleation (Paoli et al., 2004; Jensen and Pfister, 2004; Shirgaonkar and Lele, 2006; Sölch and Kärcher, 2010), and were later extended to study aerosol indirect effects with a superdroplet approach (Andrejczuk et al., 2008) in which each "superdroplet" represents a mutliplicity of modeled particles with identical at-

tributes, such as size and chemical properties. Later, the SDM was extended to include a stochastic representation of collisional coalescence (Shima et al., 2009; Riechelmann et al., 2012) and ice-phase processes (Shima et al., 2020), making the SDM a nearly-complete Monte Carlo representation of cloud microphysics. The burgeoning field of particle-based cloud microphysics uses SDM implementations in large-eddy simulations (LES) to understand microphysical processes that are underresolved in traditional bulk and bin methods (e.g., Chandrakar et al., 2021; Andrejczuk et al., 2010; Morrison et al., 2019; Dziekan et al., 2019; Grabowski, 2020; Hoffmann, 2017). Furthermore, a growing literature of machine learning in microphysics utilizes the SDM as a source of high-fidelity training data from which to "learn" microphysical tendencies and properties (Bieli et al., 2022; Seifert and Rasp, 2020). However, without a complete representation of microphysical processes in the SDM, its predictive and benchmarking power for cloud feedbacks is limited.

Many implementations of the SDM do not include the process of collisional breakup of droplets. Not only is collisional breakup a highly uncertain process in existing bin and bulk parameterizations (Morrison et al., 2020; Grabowski et al., 2019), but these uncertainties have been found to impact rain rates and other macroscale quantities in bin microphysics studies (Seifert et al., 2005). Studying collisional breakup in the SDM is not straightforward, as it requires balancing computational complexity within a mass-conserving numerical model that respects the physics of the process. Notably, a single breakup event is likely to produce fragments of multiple different sizes. A literal representation of all fragments in the SDM would require the creation of new superdroplet tracers for each new droplet size resulting from breakup, which can lead to an explosive growth of super-droplet quantity and dramatically inhibit performance of the SDM. Another option that reduces the computational burden of new superdroplets involves creating and then selectively merging superdroplets, as in Bringi et al. (2020). While both options respect mass conservation and the physics of superdroplet breakup, a scalable adaption of the SDM for parallel applications such as LES requires strict conservation of the total number of superdroplets. This work proposes a superdroplet-conserving SDM algorithm for the process of collisional breakup, conceptually similar to the mass-flux algorithm of Kotalczyk et al. (2017), using a Monte Carlo step that samples from a fragment size distribution.

This superdroplet-conserving breakup implementation draws inspiration from an analogous "superparticle" representation of phytoplankton (Jokulsdottir and Archer, 2016): individual phytoplankton aggregates spontaneously break uniformly into a number of fragments determined by a power law probability distribution. We apply a similar spontaneous breakup principle to an intermediate coalesced state resulting from the collision of two droplets. (While spontaneous breakup of liquid water droplets has also been investigated (Kamra et al., 1991), it has not been observed in in-situ studies of droplet collisions (Testik and Rahman, 2017) and is not included in this version of the SDM.) The presented collisional breakup algorithm utilizes empirical collection/breakup efficiencies (such as Schlottke et al. (2010); Beard and Ochs (1995); Berry (1967)) to determine whether a colliding droplet pair is likely to break-up, and then samples from a corresponding empirical fragment size distribution (such as Low and List (1982); Schlottke et al. (2010); Beard and Ochs (1995); McFarquhar (2004)) to determine the properties of the resulting fragmented superdroplet. Breakup parameterizations are typically very complex and aim to summarize multiple physical mechanisms of breakup. This work addresses how the proposed SDM breakup algorithm samples from such complex fragment size distributions, but leaves evaluation and analysis of these empirical distributions to future work.

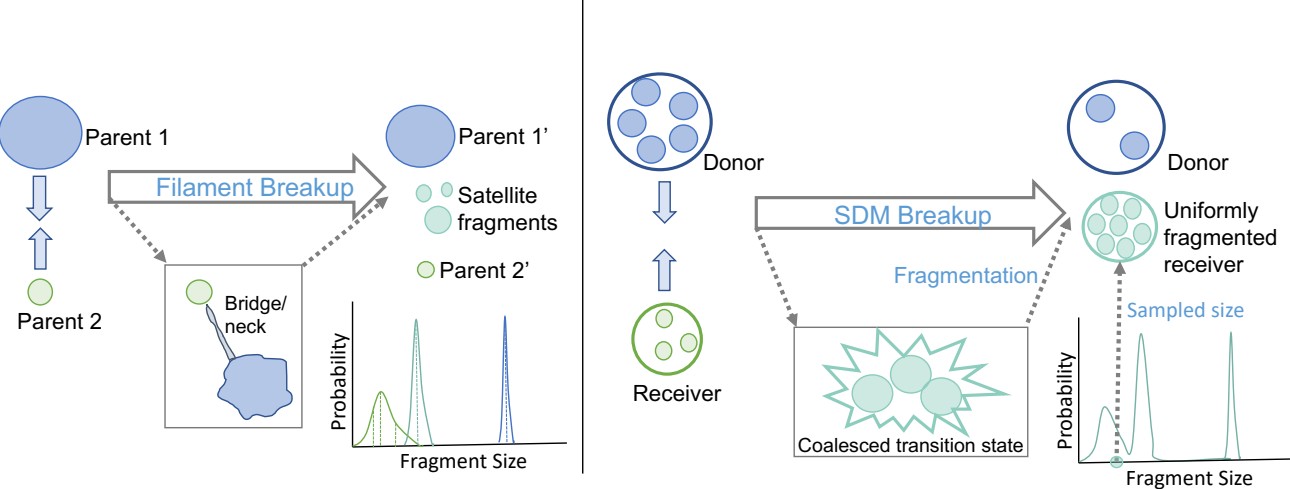

**Figure 1.** Conceptual view of a real filament breakup event (left) and the superdroplet-number-conserving SDM collisional breakup dynamic described in this work (right). The real event involves collision between two parent droplets, which may form a neck bridging each other before fragmenting into several differently sized droplets. The resulting droplets consisting of larger fragments derived from the initial colliding parents, plus a set of small fragments known as "satellites." Each of these groups forms a subdistribution in the overall fragmentation function, shown below the droplet schematics. The tracer-conserving SDM representation involves collision of two groups of droplets (each group represented as one superdroplet, a donor and receiver), which collide and coalescence into a transition state, which then fragments uniformly to a size sampled from the same fragmentation function as in the real case. The result of the SDM breakup is two superdroplets, or two groups of droplets, with one group corresponding to leftover donor droplets, and the other group corresponding to a set of fragments whose size may correspond to the depleted parent droplet size or the satellite fragments, depending on the fragment size sampling step.

The contents of this paper proceed as follows: Section 2 begin with a conceptual description of the proposed breakup algorithm, followed by a mathematical description of its implementation. Section 3 validates the implementation against analytical results in the literature and explores convergence with number of superdroplets. Section 4 then presents several idealized simulations including various parameterizations of collisional breakup to demonstrate the behavior of this implementation in the SDM. Section 5 concludes the discussion and poses additional scientific questions which may be within reach given this novel
implementation.

## 2    Superdroplet-conserving Collisional Breakup

### 2.1    Conceptual description

Two colliding liquid hydrometeors in the atmosphere can break-up via several physical pathways, including filament, sheet, and disc breakup (Barros et al., 2008). The colliding droplets, referred to as "parents", typically lose mass to newly-formed tiny
"satellite" droplets that result from the collision, thereby resulting in several differently sized droplet fragments (see Figure 1,

left). As noted previously, computational scaling of the SDM relies on preserving the number of tracers in the system. In order to preserve the number of superdroplets in a binary collisional breakup event, breakup is treated as a two step process based on superdroplet-conserving coalescence (Figure 1, right). First, the two superdroplets collide and coalesce: the superdroplet of higher multiplicity acts as a "donor" by donating mass and multiplicity while maintaining its attributes; the other superdroplet

acts as a "receiver" by growing in mass and maintaining its multiplicity to form a "coalesced transition state." This unstable coalesced transition state immediately breaks up into fragments of uniform size: the fragment size is selected sampling from a distribution of fragment sizes that encompasses both the remnants of the original parent droplets, as well as the distribution of satellite fragments that can result from the collision. (Going forward, the term "fragment" will be used to describe all collisional breakup products, both small satellite fragments as well as larger fragments that are nearer in size to the original

colliding droplets.) The attributes and multiplicity of this fragmented receiver are updated, with multiplicity increasing and mass of the individual droplets represented by the superdroplet decreasing. Uniform fragmentation is required to maintain conservation of superdroplets. Furthermore, uniform fragmentation requires the assumption that all superparticle attributes are extensive quantities and undergo equipartitional splitting (not applicable, e.g., for insoluble aerosol constituents). The product of a collisional breakup event is therefore two superdroplets: the donor maintains its attributes but donates multiplicity, and the

fragmented receiver represents (uniform) fragments that result from the breakup event following a coalesced transition state. As in the original Monte Carlo step that determines whether a collision occurs, the fragment size is sampled at random from a fragment size distribution, which may depend on the properties of the colliding particles.

## 2.2 Mathematical description

The superdroplet-conserving method of collisional breakup is illustrated in Figure 2 and formulated below using notation

following work of Shima et al.. A single superdroplet with label $i$ has a position $\mathbf{x}_i(t)$ and extensive physical attributes $\mathbf{a}_i(t)$, such as droplet volume or mass $(v_i(t), m_i(t))$ or mass of solute $(M_i(t))$. (Note that in Shima et al.'s notation, $\mathbf{a}_i(t)$ includes attributes such as droplet radius, whereas we only consider attributes which are linearly additive and extensive in the droplet size, such as volume or mass.) For simplicity, we will generally group all such extensive attributes together as $\mathbf{a}_i(t)$, but will specifically use the droplet mass $m_i$ in computations of the transfer of extensive properties between superdroplets. Each

superdroplet corresponds to a multiplicity $\xi_i(t)$ of "real" droplets which exist in the same gridbox and have identical such attributes.

The proposed breakup algorithm unifies the representation of collisional coalescence and breakup and builds on the original coalescence Monte Carlo steps in Shima et al.. As in this original SDM, we begin by selecting pairs of superdroplets to consider collisions:

1. All superdroplets within a cell are randomly ordered in a list of non-overlapping pairs $(j_\alpha, k_\alpha)$ where $j$ and $k$ are the superdroplet indices, and $\alpha$ refers to the pair index.

Next, we determine how many collisions, $\gamma_\alpha$, occur for the pair $\alpha$ in the time step:

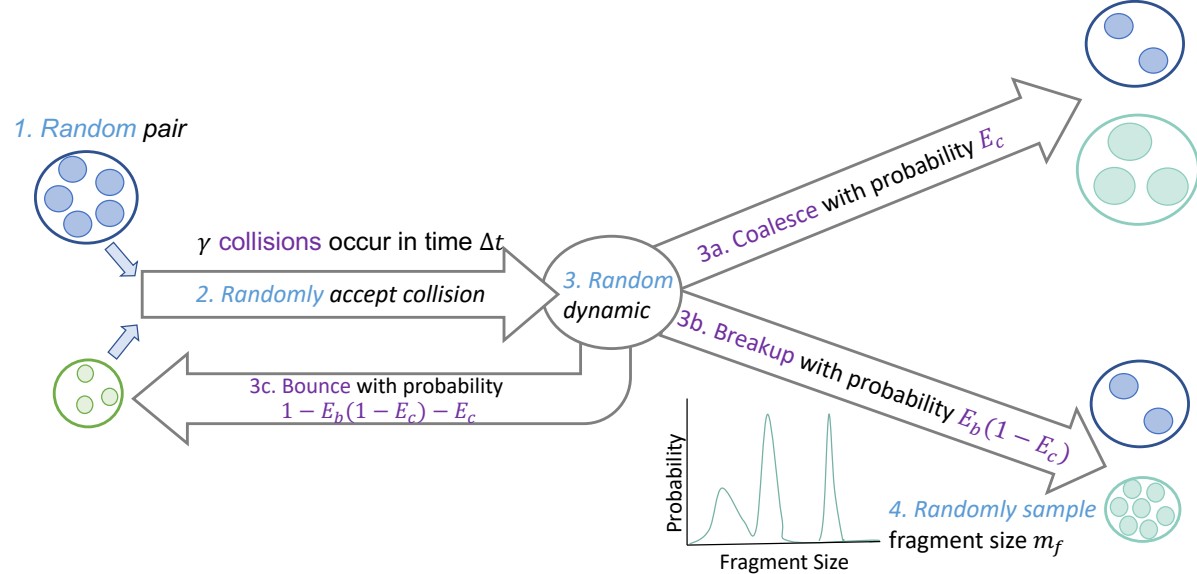

**Figure 2.** Diagram of the Monte Carlo decision pathway during a collision-coalescence-breakup event in the proposed algorithm.

2. The probability of collision between droplets $i$ and $j$ is given by

$$P_{i,j} = K_{i,j}\Delta t \tag{1}$$

where $K_{i,j}$ is the rate of collisions based on the properties of droplets $i$ and $j$, and $\Delta t$ is the model time step. The scaled probability of collision $P_\alpha^{(s)}$ for this pair $\alpha$ accounts for the multiplicities of the colliding superdroplets:

$$P_\alpha^{(s)} = \max(\xi_j, \xi_k)P_\alpha. \tag{2}$$

Only a subset $\lfloor n_s/2 \rfloor$ of possible SD pairs are considered out of all possible superdroplet pairs at each time step. Therefore, the probability is further scaled up to form the corrected probability of collision:

$$p_\alpha = \frac{n_s(n_s-1)}{2}\bigg/\left\lfloor\frac{n_s}{2}\right\rfloor P_\alpha^{(s)}. \tag{3}$$

The number of collisions that occur in this time step, $\gamma_\alpha$, is then determined in a Monte Carlo step based on $p_\alpha$. Taking $\phi_\alpha \in (0,1)$ to be a uniform random number,

$$\gamma_\alpha = \min(\lceil p_\alpha - \phi_\alpha \rceil, \lfloor \xi_{j_\alpha}/\xi_{k_\alpha} \rfloor). \tag{4}$$

Here, we assumed the superdroplets are ordered such that $\xi_{j_\alpha} \geq \xi_{k_\alpha}$. If $\gamma_\alpha = 0$, then no collisions occur.

The collision rate is then $\gamma_\alpha$ collisions per gridbox and per time step. Due to the constraint in equation (4) based on droplet multiplicity, some collisions which should occur probabilistically cannot if the donor superdroplet has insufficient multiplicity

to collide $p_\alpha$ times. Therefore, a collision deficit $p_\alpha - \gamma_\alpha$ may be tracked as a tool to assess whether the model time step is sufficiently small (elimination of the the collision deficit is used for adaptive step size control in the SDM implementation used herein (Arabas et al., 2022)).

In the original SDM, particles coalesce as long as $\gamma_\alpha > 0$, as the rate of collisions is taken to refer only to collisional coalescence. However, when we consider collisional breakup, an additional Monte Carlo step must be taken to determine whether the particles coalesce or break up. This is determined based on a coalescence efficiency (or collection efficiency) $E_c$, which generally depends on properties of the colliding particles such as their fall speed, mass, and surface tension. We additionally account for the fact that in some collisions, droplets may bounce off of one another elastically by including an
optional additional parameter for the breakup efficiency, $E_b$. This second Monte Carlo step is summarized as follows.

3. Compute the dynamic that occurs: coalescence, breakup, or bounce (nothing). A second uniform random number $\phi'_\alpha$ determines the outcome:

$$\text{dynamic}_\alpha = \begin{cases} \text{coalescence,} & \phi'_\alpha \leq E_c(\mathbf{a}_j, \mathbf{a}_k) \\ \text{breakup,} & E_c(\mathbf{a}_j, \mathbf{a}_k) < \phi'_\alpha \leq E_b(\mathbf{a}_j, \mathbf{a}_k)(1 - E_c(\mathbf{a}_j, \mathbf{a}_k)) + E_c(\mathbf{a}_j, \mathbf{a}_k) \\ \text{bounce,} & \phi'_\alpha > E_b(\mathbf{a}_j, \mathbf{a}_k)(1 - E_c(\mathbf{a}_j, \mathbf{a}_k)) + E_c(\mathbf{a}_j, \mathbf{a}_k) \end{cases} \tag{5}$$

Once the dynamic is determined, a fragment size is sampled if necessary:

4. Sample a fragment size $m_{f,\alpha}$ (mass) from a fragment size distribution, $P_{f,\alpha}$, with cumulative distribution function (CDF) $C_{f,\alpha}(\phi)$ that depends on the colliding particle attributes. A related variable, $N_{f,\alpha}$, is taken to denote the number
of fragments that would form in a collision between droplets of mass $m_j$ and $m_k$: $N_{f,\alpha} = \frac{m_k + m_j}{m_{f,\alpha}}$.

Finally, updating of multiplicities and attributes proceeds based on the selected dynamic, number of collisions, and sampled fragment size (if applicable):

(a) For coalescence:

$$\begin{cases} \xi'_j = \xi_j - \gamma_\alpha \xi_k \\ \mathbf{a}'_k = \mathbf{a}_k + \gamma_\alpha \mathbf{a}_j \\ \text{if } \xi'_j = 0, \text{ then } \xi'_j, \xi'_k = \xi_k/2, \ \mathbf{a}'_j = \mathbf{a}_k \end{cases} \tag{6}$$

The coalescence rate is incremented by $\gamma_\alpha \xi_k$. The final step in updating multiplicities and attributes serves to conserve the number of superdroplets with nonzero multiplicity in the simulation in the case that all droplets within superdroplet $j$ are depleted. Unlike in Shima et al. (2009), where superdroplet $j$ is discarded, this approach sets properties of superdroplet $j$ to be identical to $k$, and both $j$ and $k$ to half of $k$'s multiplicity to conserve mass.

(b) For breakup:

In some cases, only $\gamma_{jk} \leq \gamma_\alpha$ breakups can occur for a given superdroplet pair without encountering negative multiplicities. We compute this maximum possible number of breakup steps and update the superdroplet properties using a recurrence relation (assuming $\gamma_\alpha > 0$), and track a breakup deficit rate of $\gamma_\alpha - \gamma_{jk}$. (Alternatively, one may perform substepping of the breakup event.) The particle attributes are updated such to be consistent with the result of several breakup steps with $\gamma_\alpha = 1$ occurring in sequence, always producing fragments of size $m_{f,\alpha}$.

$$
\begin{cases}
\gamma_{jk} = 0 \\
\xi_j^{\text{transfer}} = 0, \quad \xi_{j,\text{next}}^{\text{transfer}} = \xi_k \\
\xi_k^{\text{new}} = \xi_k, \quad \xi_{k,\text{next}}^{\text{new}} = \xi_k \left( \frac{m_j + m_k}{m_{f,\alpha}} \right) \\
\text{while } \gamma_{jk} < \gamma_\alpha \text{ and } \xi_{j,\text{next}}^{\text{transfer}} \leq \xi_j : \\
\qquad \xi_j^{\text{transfer}} = \xi_{j,\text{next}}^{\text{transfer}} \\
\qquad \xi_k^{\text{new}} = \xi_{k,\text{next}}^{\text{new}} \\
\qquad \gamma_{jk} = \gamma_{jk} + 1 \\
\qquad \xi_{j,\text{next}}^{\text{transfer}} = \xi_{j,\text{next}}^{\text{transfer}} + \xi_{k,\text{next}}^{\text{new}} \\
\qquad \xi_{k,\text{next}}^{\text{new}} = \xi_{k,\text{next}}^{\text{new}} \left( \frac{m_j}{m_{f,\alpha}} \right) + \xi_{k,\text{next}}^{\text{new}}
\end{cases}
\tag{7}
$$

$$
\begin{cases}
\xi_j' = \xi_j - \xi_j^{\text{transfer}} \\
\xi_k' = \xi_k^{\text{new}} \\
\mathbf{a}_k' = \frac{\xi_k \mathbf{a}_k + \xi_j^{\text{transfer}} \mathbf{a}_j}{\xi_k^{\text{new}}} \\
\text{if } \xi_j' = 0, \text{ then } \xi_j', \xi_k' = \xi_k^{\text{new}}/2, \mathbf{a}_j' = \mathbf{a}_k'
\end{cases}
\tag{8}
$$

The breakup rate is incremented by $\gamma_{jk}\xi_k$. The breakup deficit rate is incremented by $(\gamma_\alpha - \gamma_{jk})\xi_k$

(c) For bounce:

No update is made to droplet multiplicities or attributes, and only the collision counter is incremented.

## 2.3 Additional Implementation Details

This method of breakup allows for the splitting of a coalesced transition state into a non-integer number of fragments ($N_{f,\alpha} = \frac{m_k + m_j}{m_{f,\alpha}}$ need not be integer), depending on the sampled fragment size. For instances where it may be desirable to preserve superdroplet multiplicities as integers, we recommend rescaling the multiplicities after the breakup step by a factor of $r_k = \lceil \xi_k \rceil / \xi_k$, and the multiplicities correspondingly by $1/r_k$ such that extensive attributes (including mass) are conserved.

The presence of a "breakup deficit" in the case where $\gamma_{jk} < \gamma_\alpha$ can be averted by substepping, though this is inadvisable for highly parallel applications of the SDM. Furthermore, superdroplet multiplicities may increase without bound according to the algorithm as presented above, which can lead to numerical artifacts and instability within a simulation. A set of limiters preventing runaway multiplicity is discussed in Appendix A. Finally, a method for sampling a fragment size from a highly nonlinear empirical distribution, such as Straub 2010, is discussed in Appendix B.

## 3   Validation and Convergence Properties


    In order to validate the proposed Monte Carlo algorithm which encompasses both coalescence and breakup, we compare to an analytical solution to the generalized stochastic collection equation (SCE, or Smoluchowski equation). The approach is similar to that of Lee and Matsoukas (2000) and compares to the solution of Srivastava (1982), which uses constant-in-time and attribute-independent coagulation and breakup rates. Discussion of the solution outside of cloud-physics can be found,

e.g., in Hansen (2018, eq. 8.58 therein) where it is presented in the context of polymerization-depolymerization modeling (see Blatz and Tobolsky, 1945, for a relevant seminal work featuring analytic SCE solutions). Specifically, herein we compare to the results including processes of binary coagulation and breakup only, neglecting spontaneous fragmentation.

    The deterministic solution to the SCE relates the evolution in time of the ratio $m$ of mean mass of particles to the fragment mass $m_{\text{frag}}$ in the case of constant-coefficient binary coagulation and breakup is given by (notation as in eq. 13 in Srivastava,

150  1982):

$$m(\tau) = m(0)e^{-\beta_\star \tau} + \left(1 + \frac{1}{2\beta_\star}\right)\left(1 - e^{-\beta_\star \tau}\right). \tag{9}$$

The solution is given using non-dimensional variables defined by:

$$\tau \;\;=\;\; cMt \tag{10}$$

$$\beta_* \;\;=\;\; \beta/c \tag{11}$$

where $t$ is time, $c$ is a constant coalescence rate, $M$ is the ratio of total mass of the system to the fragment mass $m_{\text{frag}}$ and $\beta$ is a constant breakup rate. The rates $c$ and $\beta$ correspond to efficiencies $E_c = c/(c+\beta)$ and $E_b = 1$ (no bouncing) with the corresponding collision kernel $K = c + \beta$. Of note, the solution does not depend on the initial particle size spectrum – only on the initial mean mass $m(0)$.

    Two caveats are involved in comparing SDM results against SCE solutions in this test set up. First, the constant collision

kernel admits collisions of same-sized particles, whereas collisions with a single superdroplet are not included in the present SDM implementation. This discrepancy should diminish with increasing numbers of super-particles (and hence decreasing values of multiplicities) down to zero for a one-to-one simulation with multiplicities of unity. The second caveat involves the assumption of breakup resulting in a only a single fragment size, $m_{\text{frag}}$. This simplification is required to attain the analytic solutions of Srivastava (1982), but removes the final of four stochastic elements in the proposed superdroplet breakup algorithm.

As such, the following comparisons are useful to understand the convergence properties of the proposed algorithm's first three stochastic elements (superdroplet sampling, collision probability, coalescence probability) only.

| Case | $c$ (s$^{-1}$) | $\beta$ (s$^{-1}$) | $m_{\text{frag}}$ (g) |
|------|------|------|------|
| breakup-only | $10^{-15}$ | $10^{-9}$ | 0.25 |
| coalescence-only | $0.5 \times 10^{-6}$ | $10^{-15}$ | - |
| coalescence-breakup | $0.5 \times 10^{-6}$ | $10^{-9}$ | 0.25 |

**Table 1.** Coalescence rate ($c$), breakup rate ($\beta$), and fragment mass values for different simulation setups. The inactive process rates are set to $10^{-15}(s^{-1})$ rather than exactly 0 in accordance with the solutions of Srivastava (1982).

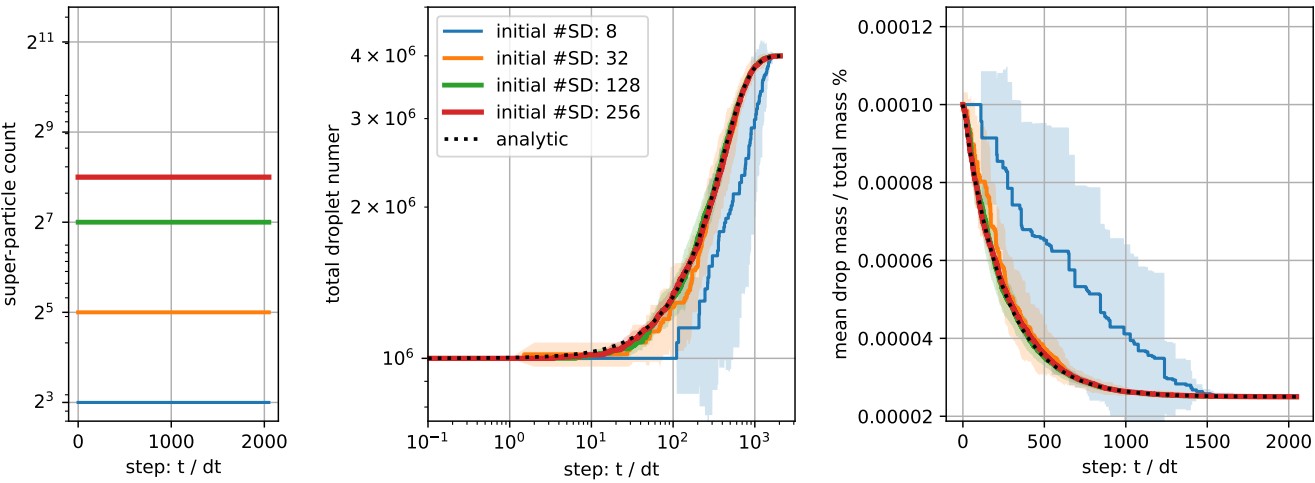

**Figure 3.** Mean (solid line) and standard deviation (shading) for the time evolution of the breakup-only dynamics, including the analytical solution of Srivastava (1982) and the SDM using 8, 32, 128, or 256 superdroplets.

We analyze three comparison cases in a zero-dimensional box setting: breakup only ($c \approx 0$), coalescence only ($\beta \approx 0$), and coalescence plus breakup. The parameter values defining the three aforementioned cases are summarized in Table 1. These simulations utilize the open-source Pythonic superdroplet code 'PySDM' (Bartman et al., 2022b; de Jong et al., 2023). For each test case, we perform the simulation at a few resolutions (number of superdroplets). Note while the SDM implementation guarantees that the number of super-particles cannot increase, a superdroplet may be removed from the system during a collision event (for details see point (5) in section 5.1.3 in Shima et al. (2009)). The simulations are performed for 2048s with 1s timesteps with adaptive collision substepping enabled. The initial size distribution is monodisperse with equal multiplicities for all superdroplets. These settings correspond to a population of $10^6$ particles in one cubic metre, with each droplet having initial mass of 1g. In the solution of Srivastava (1982), it is assumed that all fragments resulting from breakup are of equal size. Here, the constant fragment mass is set (arbitrarily) to 0.25g.

Figures 3–5 present the mean and standard deviation of 10 realizations of the SDM simulations, versus the analytic solutions. Both the breakup-only (Figure 3) and breakup-plus-coalescence (Figure 5) cases feature asymptotic values of droplet

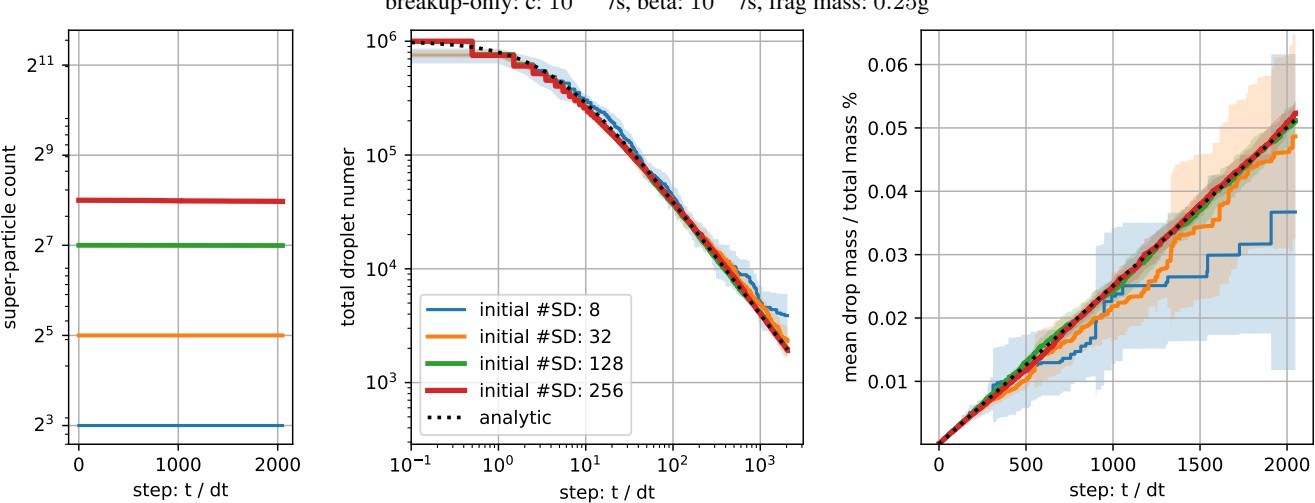

**Figure 4.** Mean (solid line) and standard deviation (shading) for the time evolution of the coalescence-only dynamics, including the analytical solution of Srivastava (1982) and the SDM using 8, 32, 128, or 256 superdroplets.

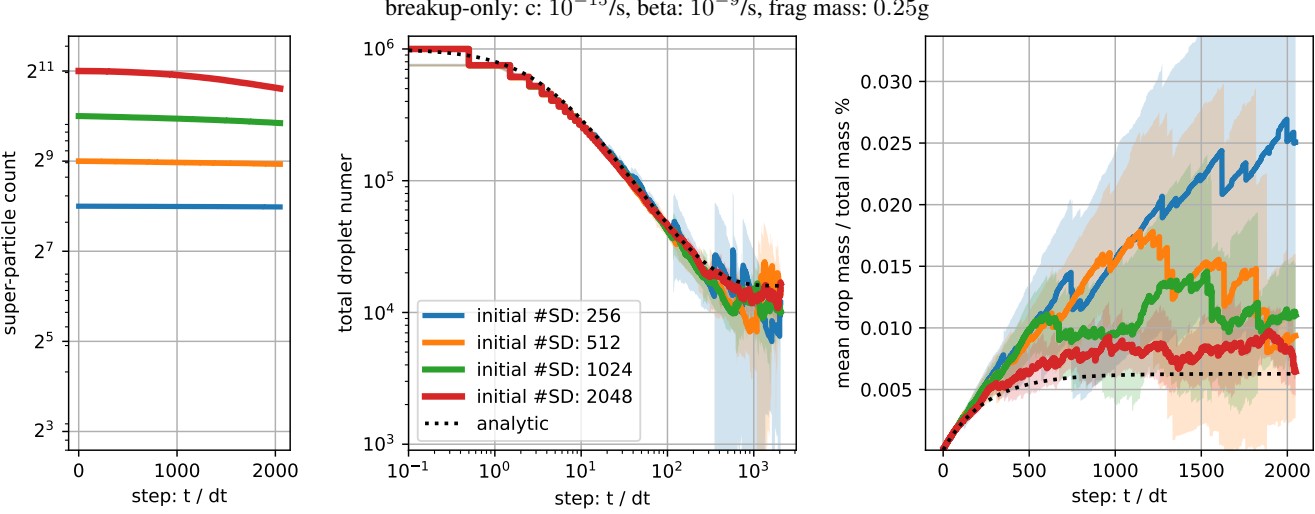

**Figure 5.** Mean (solid line) and standard deviation (shading) for the time evolution of the coalescence-breakup combined dynamics, including the analytical solution of Srivastava (1982) and the SDM using 256, 512, 1024, or 2048 superdroplets.

number and mass, corresponding to all droplets having the size $m_\text{frag}$ (breakup-only) or a balance between fragmentation and
coalescence of droplets (breakup-plus-coalescence). In all three sets of dynamics, increasing the number of superdroplets robustly reduces the ensemble spread and improves the match with the analytic solution. Much higher resolution (on the order of thousands superparticles) is required to match the analytic solution for the breakup-coalescence equilibrium than for simulations featuring each process separately. This discrepancy is related to the additional stochastic step of selecting the breakup or coalescence dynamic. Extension of this logic implies that even more superparticles would be required to match the exact
asymptotic behavior of a system with a distribution of fragment sizes, due to yet an additional stochastic element of sampling.

In Figure 5, the simulation with the largest number of superdroplets displays removal of superdroplets from the system. The removal can happen when coalescence leads to zero multiplicity in one of the resultant superdroplets, which then cannot be split into two superdroplets. This is more likely when multiplicities are on average lower (i.e. when more superdroplets are used). In the limit of very few superdroplets, both the breakup-only and the coalescence-only simulations are characterized
by underestimation of the process rate, which is consistent with the SDM implementation that neglects collisional dynamics within a superdroplet.

## 4 Numerical Experiments and Discussion

To demonstrate the behavior of a particle population under the proposed breakup algorithm, we focus here on sensitivity studies in a zero-dimensional box setting and in a one-dimensional rainshaft setup. The simulations presented in this section
use a geometric collision kernel, where the rate of collisions $K_{jk}$ between droplets with the properties of superdroplets $j$ and $k$ is given by

$$K_{jk} = \pi(R_j + R_k)^2|v_j - v_k| \tag{12}$$

where $R_j$ is the radius of particle $j$ and $v_j$ is the terminal velocity/fall speed of particle $j$, computed using the parameterization of Gunn and Kinzer (1949). As in Shima et al. (2009), collisions within a superdroplet (i.e. collisions between droplets
represented by the same superdroplet) are not considered – in line with the use of a geometric collision kernel which precludes collisions between equally-sized droplets, as they have the same terminal velocity. (These intra-superdroplet collisions may be important in the case of turbulent collision kernels.)

The coalescence efficiency is specified to be either a constant value (for sensitivity studies), or the empirical coalescence efficiency of Straub et al. (2010) which depends on the Weber number of the colliding droplet pair. (The Weber number is a
ratio of kinetic collisional energy and surface tension, and relates to the stability of a droplet pair under collision.) We consider three types of fragmentation functions: (1) a constant fragment number $N_f$, in which the particle-size distribution (PSD) is a delta function $P_f(m_{f,\alpha}) = \delta(m_{f,\alpha} - \frac{m_j + m_k}{N_f})$; (2) an exponential distribution $P_f(m_{f,\alpha}) \sim \exp(-m_f/\mu)$ where the scale $\mu$ is specified; and (3) the empirically derived fragmentation function of Straub et al., which uses four modes of fragmentation represented by lognormal or normal subdistributions.

## 4.1 Particle Size Distribution

The zero-dimensional box simulations include collisional-coalescence and collisional-breakup dynamics only. The droplet size distribution is initialized to an exponential distribution in mass $x$, given by $N(x) = x_0 \exp(-x/x_0)$ with the characteristic size $x_0 = (4\pi/3)R_0^3$ set using $R_0 = 30.531\mu$m as in Shima et al. (2009). The simulations employ $2^{13} = 8192$ superdroplets to represent a number density of $100\text{cm}^{-3}$ in a box of volume $1\text{m}^3$ with a fixed time step of $1$s. This choice of superdroplet quantity is sufficient to produce consistent results in the PSD across realizations using a different random seed, and was shown by Shima et al. to closely match the exact PSD in a similar box model simulation of collisional coalescence.

Particle size distributions are displayed as the number distribution or as the marginal mass distribution $g(R) = \frac{dm}{dln(R)} = 3x^2 n(x)$ where $n(x)$ is the particle size distribution. This mass distribution is computed by binning the resulting superdroplets into 128 logarithmically-spaced size bins between particle radius 1μm and radius 10mm. We separate the simulations into those which use a deterministic fragmentation function, in which breakups result in a constant number of fragments in any given collision; a stochastic fragmentation function with fragment sizes sampled from a specific distribution; and a size-dependent fragmentation function, where the fragment sizes are sampled from a distribution whose parameters depend on the colliding particles. We further include experiments exploring the use of a fixed coalescence efficiency versus a particle-attribute-derived coalescence efficiency. This separation elucidates which aspects of the particle population behavior are attributable to stochastic sampling of the fragmentation function, or related to particle-property-dependent parameters such as Weber number.

### 4.1.1 Sensitivity Studies: Deterministic and Size Independent Fragmentation

First we investigate the sensitivity of the PSD evolution to the coalescence efficiency, using four values of a constant-valued efficiency $E_c$ between 0.7 and 1.0 ($E_c = 1.0$ corresponds to coalescence-only) and a particle-size dependent $E_c$ parameterization (Straub et al., 2010). All simulations use a deterministic fragmentation function in which all single-step collisional breakups result in $N_f = 8$ fragments:

$$P_{f,\alpha}(m_f) = \delta\left(m_f - \frac{m_j + m_k}{N_f}\right). \tag{13}$$

Figure 6 displays two snapshots of the PSD under this set of dynamics, demonstrating the additional Monte Carlo step of selecting whether coalescence or breakup occurs, independent of sampling a fragment size. As expected, the initial PSD broadens and shifts toward larger droplets at 100s, with the largest values of fixed $E_c$ leading to the largest increase in average particle mass. However, after 200s, the PSD for the $E_c = 0.8$ case remains approximately steady with a mean size that is smaller than the initial distribution mean, demonstrating that coalescence and breakup are approximately balanced in this case.

By contrast, the PSD for the Straub 2010 parameterization of $E_c$ initially between the $E_c = 0.9$ and $E_c = 1.0$ simulations at 100s, but narrows without shifting toward much larger droplets after further time has elapsed, leading to a dominant mode that is more similar to the $E_c = 0.9$ case. This empirical parameterization also shows evidence of approaching a steady state distribution, in which coalescence and breakup rates are matched on average, driving the PSD to a stationary state. The Straub 2010 parameterization decreases exponentially with the colliding particle Weber number, which is correlated with the size and

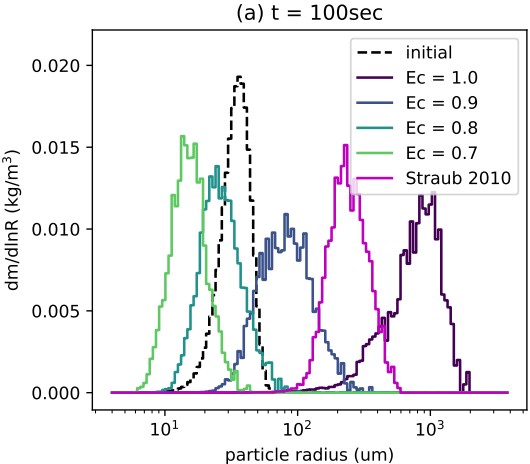
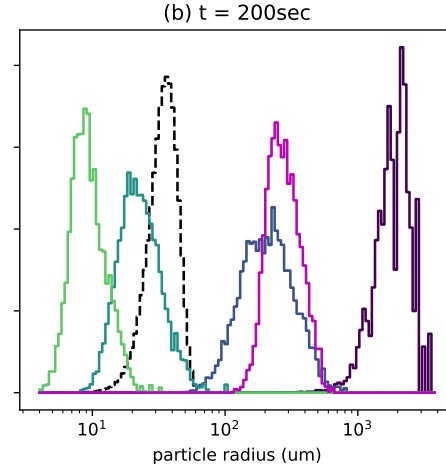

**Figure 6.** Particle size distribution with varying coalescence efficiencies under a geometric collision kernel after 100s (left) and 200s (right). The breakup fragmentation function is deterministic, with the fragment size determined as 1/8th the sum of the colliding droplet sizes. The dashed black line represents the initial PSD, and solid lines represent various fixed values of the coalescence efficiency. The pink line corresponds to a size-dependent coalescence efficiency from Straub et al..

relative terminal velocity of the colliding particles. Two colliding particles of comparable size have a low relative terminal velocity, therefore as the PSD shifts toward larger coalesced droplets, there is a competing effect between a larger particle size increasing the Weber number, and decreased relative terminal velocity reducing it. This competition produces the stationary

behavior and narrowing of the PSD observed in this case.

Next we consider the PSD evolution when the coalescence efficiency is held fixed at a constant value and the fragmentation function is varied. In Figure 7(a), we consider a deterministic fragmentation function where the number of fragments from a single breakup event is fixed (as in Figure 6), as well as an exponential fragment size distribution with scale $\mu$ specified as a multiple of the initial mean particle mass $x_0$. When the number of fragments is fixed, results using the largest number of

fragments display the smallest mean particle size and broadest spectra. The first behavior is expected, as a larger value of $N_f$ results in smaller typical fragment sizes. The broadening of the spectrum can be attributed to a wider range of collision rates between very small droplets (which result from fragmentation), and is generally an expected outcome of including collisional breakup.

When the fragment size is sampled from an exponential distribution (Figure 7(b)), the resulting spectra are bimodal, with a

large-droplet mode that larger for larger choices of the mean fragment size $\mu$, and a narrow small-droplet mode that likewise depends on $\mu$. The appearance of a second mode occurs when the fragment size is sampled from the left tail of the fragment size distribution, whereas the large mode corresponds to a droplets undergoing coalescence only, as in the $N_f = 1$ case. This behavior indicates that through stochastic sampling of the dynamic and fragment size together, the droplet population splits into one mode which fragments into smaller droplets, and a second mode which primarily undergoes coalescence and grows in

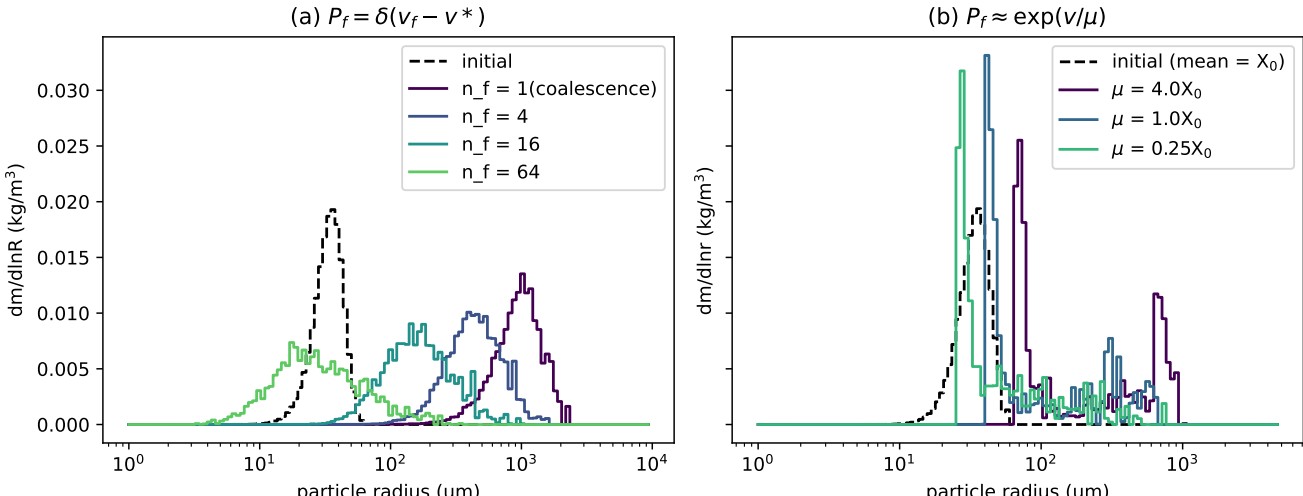

**Figure 7.** Sensitivity to fragmentation function of PSDs following collisions with a geometric kernel and fixed coalescence efficiency of $E_c = 0.95$ in a deterministic and stochastic fragmentation function case. (Left) The fragment size is fixed by a divisor of the sum of colliding particle volumes; (right) fragment size is sampled from an exponential distribution with varying means $\mu$ determined as a multiple of the initial distribution mean. The initial distribution is shown as a black dashed line in each figure.

size. Because larger droplets collide at much quicker rates than small droplets, the fragmented mode is less likely to collide and re-coalesce to form medium-sized droplets, while the coalesced-mode retains some probability of colliding and either growing (coalescing), or breaking up into smaller droplets. Thus we observe that the small-droplet-mode grows in this instance, with particles effectively become "stuck" in this dynamical regime due to the separation of scales in collision rates.

### 4.1.2   Steady State under Stochastic Size-Dependent Fragmentation

Finally, we consider an empirically derived coalescence efficiency and corresponding fragmentation function whose parameters depend on the colliding droplet properties (Straub et al., 2010). In Figure 8, we consider the evolution of the PSD under the Straub 2010 efficiency and fragmentation dynamics, beginning from the same initial distribution as previous experiments. At first, the PSD broadens and shifts towards larger droplet sizes, as in Figure 6, but the PSD after 7200s shows little difference from the PSD at 1800s. These results indicate the stationarity of the particle size distribution after sufficient time has

elapsed: coalescence and breakup are balanced, as in the previous example. Contrasted with Figure 6, which used a deterministic size-independent fragmentation function, the stationary PSD resulting from the Straub et al. (2010) parameterization of fragmentation is less symmetric and contains multiple small peaks. This difference reflects the sampling from a multimodal distribution of fragment sizes, contrasting with the symmetric PSDs found from using a fixed number of fragments.

This empirical parameterization provides an additional opportunity for validation of the breakup algorithm on top of the

analytical results presented in Section 3. Figure 9 compares results of this SDM implementation against figure 10 of Straub

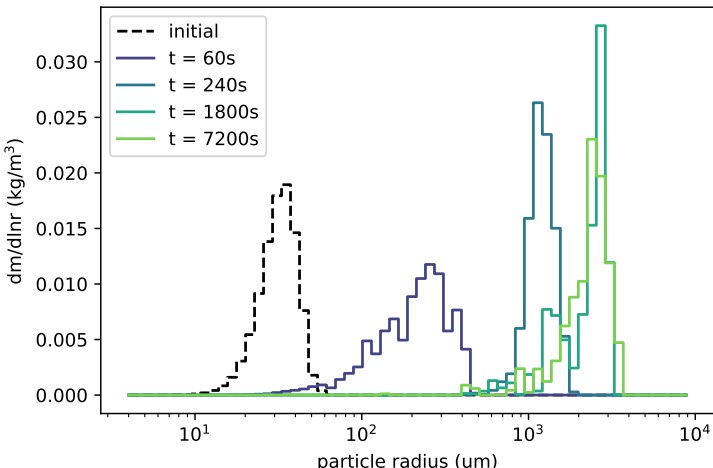

**Figure 8.** Initial PSD (black dashed) and PSD's following collisions with a geometric kernel, the Straub 2010 collection efficiency, and the Straub 2010 fragmentation function (Straub et al., 2010) after several elapsed times (colors).

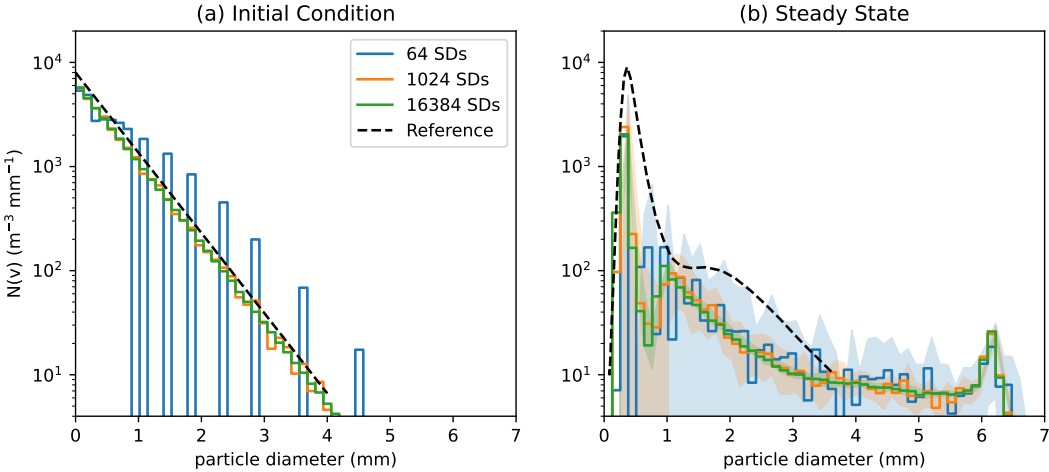

**Figure 9.** Initial PSD (left) and steady-state PSD (right) under collisions with the Straub 2010 parameterizations, including reproduction of the findings in figure 10 from Straub et al. (2010) (black dashed line), and simulations including the proposed breakup algorithm with different numbers of superdroplets. Steady state results are displayed as the mean plus or minus one standard deviation (shading) of 10 simulation instances.

et al. (2010) (note that we now plot number concentration rather than mass concentration with a logarithmic y-scale and linear x-scale, as in Straub et al. (2010) and McFarquhar (2004)). Particles are initialized as a Marshall Palmer distribution (exponential in droplet diameter) with rain rate $54\text{mmhr}^{-1}$; superdroplet sizes are sampled logarithmically over this diameter range, rather than using constant multiplicity. With only 64 superdroplets, the SDM struggles to adequately sample the low concentration but important rain droplets with diameters exceeding 1 mm, while using over 1000 superdroplets is more than sufficient. The simulation is run using a 1s time step for 7200s to approximate a steady state in Figure 9b. The SDM appears to converge using between 1000 to 16,000 superdroplets, with more superdroplets being necessary to reduce spread of the results for small particle diameters. The SDM approach captures the size and amplitude of the first mode of the steady state distribution reasonably well, but shows some discrepancies at at intermediate particle sizes, predicting a secondary mode at 1.2mm diameter rather than the less pronounced shoulder at 2mm. Inspecting the resulting distributions at large particle sizes reveals a larger quantity of 6-7mm diameter particles in the SDM simulations, which are not seen in the work of Straub et al. (2010). This behavior indicates that some large superdroplets coalesce rather than breaking up, which may be reflective of the multiplicity-limiter in the model which performs coalescence rather than breakups which would result in a very large increase in multiplicity (i.e. sampling a small fragment size during collision involving one or more large droplets). The coalescence efficiency tends toward zero for any droplet larger than 6 mm in diameter, which explains why the large droplet mass is concentrated at this size. In a realistic setting, however, most rain-range droplets would be expected to sediment before attaining this large 6mm size through coalescence. Indeed, we note that later results presented in Figure 11 do not show significant mass concentration at droplet sizes larger than 1 mm radius, indicating that this error in the steady state does not present in the transient rainshaft simulation.

## 4.2 Cloud and Precipitation Properties

Next we consider the impact of collisional breakup in a one-dimensional warm rain setting that includes condensation/evaporation (including aerosol activation/deactivation), collisions, and transport of particles within the column through advection and sedimentation/precipitation. These 1D simulations are based on the kinematic framework of Shipway and Hill, using a fixed profile of dry-air potential temperature and dry-air density $\rho_d(z)$, and a resolved budget of water vapor (advection and coupling with vapor uptake and release by particles). The vapor advection is solved using the MPDATA algorithm on a columnar grid with vertical spacing of 100m (employing the PyMPDATA implementation Bartman et al., 2022a). An aerosol population with hygroscopicity $\kappa = 0.9$ is initialized throughout the vertical domain with $2^8 = 256$ superdroplets per gridbox. This choice of 256 superdroplets per gridbox reflects the higher computational demands of the one dimensional simulation compared to the box model and still produces statistically convergent results in the macroscopic quantities investigated . For the first 600s of spin-up, condensation-evaporation (including aerosol activation using kappa-Köhler theory; implementation of these processes follows that of Arabas et al. (2015)) and particle advection with the specific updraft are the only active dynamics, with a time-varying updraft momentum flux of $\rho_d w(t) = 6\text{kgm}^{-3}\text{ms}^{-1}\sin(\pi t/600\text{s})$. After this spin-up time, the updraft velocity is set to 0, and the processes of particle displacement due to sedimentation and collision-coalescence-breakup begin. The time step is fixed at 5s throughout the simulation.

The test cases demonstrated here include a no-breakup case, a property-independent breakup case where the coalescence efficiency is fixed and fragment sizes are sampled from a fixed distribution, and the particle-property-dependent empirical coalescence efficiency and fragmentation parameterizations from Straub et al. (2010). All simulations use a geometric collision rate (equation 12) and the Gunn and Kinzer terminal velocity parameterization. In the no-breakup case, all collisions result in coalescence. In the property-independent breakup case, we fix $E_c = 0.95$ for all superdroplet collisions based on the corre-

spondence in Figure 6 to the empirical coalescence efficiency. This case samples fragment sizes from a Gaussian distribution in particle volume with mean radius 30µm and standard deviation 15µm. In contrast to the property-independent case, in which the fragmentation parameters are hand-selected, the property-dependent setting is based on empirical evidence, and is expected to be more reflective of the variability of real clouds. In both the property-independent and -dependent cases, the breakup efficiency is set to $E_b = 1$ such that all collisions result in either coalescence or breakup. To contrast the behavior of the three

cases, we consider the hydrometeor population at various altitudes throughout the simulation, as well as collision process rates and aerosol processing rates.

### 4.2.1  Hydrometeor and Cloud Quantities

The mixing ratio of cloud droplets (activated droplets of no more than 50µm radius), rain droplets (radius greater than 50µm), and the number concentration of unactivated aerosols are displayed for the three test cases in Figure 10. The no-breakup

simulation forms a cloud due to activation of aerosols between 600m and 3800m altitude until the updraft is terminated after 600s. Larger rain-range droplets form from collisional coalescence and begin to sediment out of the system in clusters, visible as distinct streaks in the $(t, z)$ plane, with surface precipitation beginning around 1100s into the simulation, depleting the cloud droplet population. The vertically-averaged particle size spectra in Figure 11 demonstrate these qualitative changes as well. At 600s, the majority of particles are micron-sized aerosols with cloud droplets beginning to form. At later times of 900s and

1200s, the particles rapidly grow to tens or hundreds of microns in size, and at 1800s, the mass distribution in the no-breakup case shows significant depletion due to precipitation of large particles.

When property-independent breakup is included, a higher concentration of cloud-sized droplets persists at cloud base, and the surface precipitation is delayed and spread out relative to the case with no breakup. This behavior indicates that rain droplets favorably break up within the cloud and especially near cloud base, fragmenting into smaller cloud droplets (the mean of the

fragment size distribution is 30µm radius, only slightly lower than the rain size range) with a lower sedimentation rate. We observe this behavior in Figure 11 as well: the property-independent case shows much smaller average particle sizes at 900 and 1200s, as well as more mass remaining in the system at 1800s due to the delayed precipitation. Furthermore, the aerosol population below cloud base is not depleted as quickly in this property-independent case, indicating a reduction in aerosol scavenging and washout that is consistent with the lower precipitation rates. These phenomena are consistent with documented

impacts of collisional breakup such as reduced surface precipitation (Seifert et al., 2005), and show that the proposed algorithm can meaningfully represent the breakup process.

The empirical property-dependent breakup case using the Straub et al. parameterizations displays hydrometeor populations that are more similar to the no-breakup case, indicating that the choice of $E_c = 0.95$ in the property-independent case likely

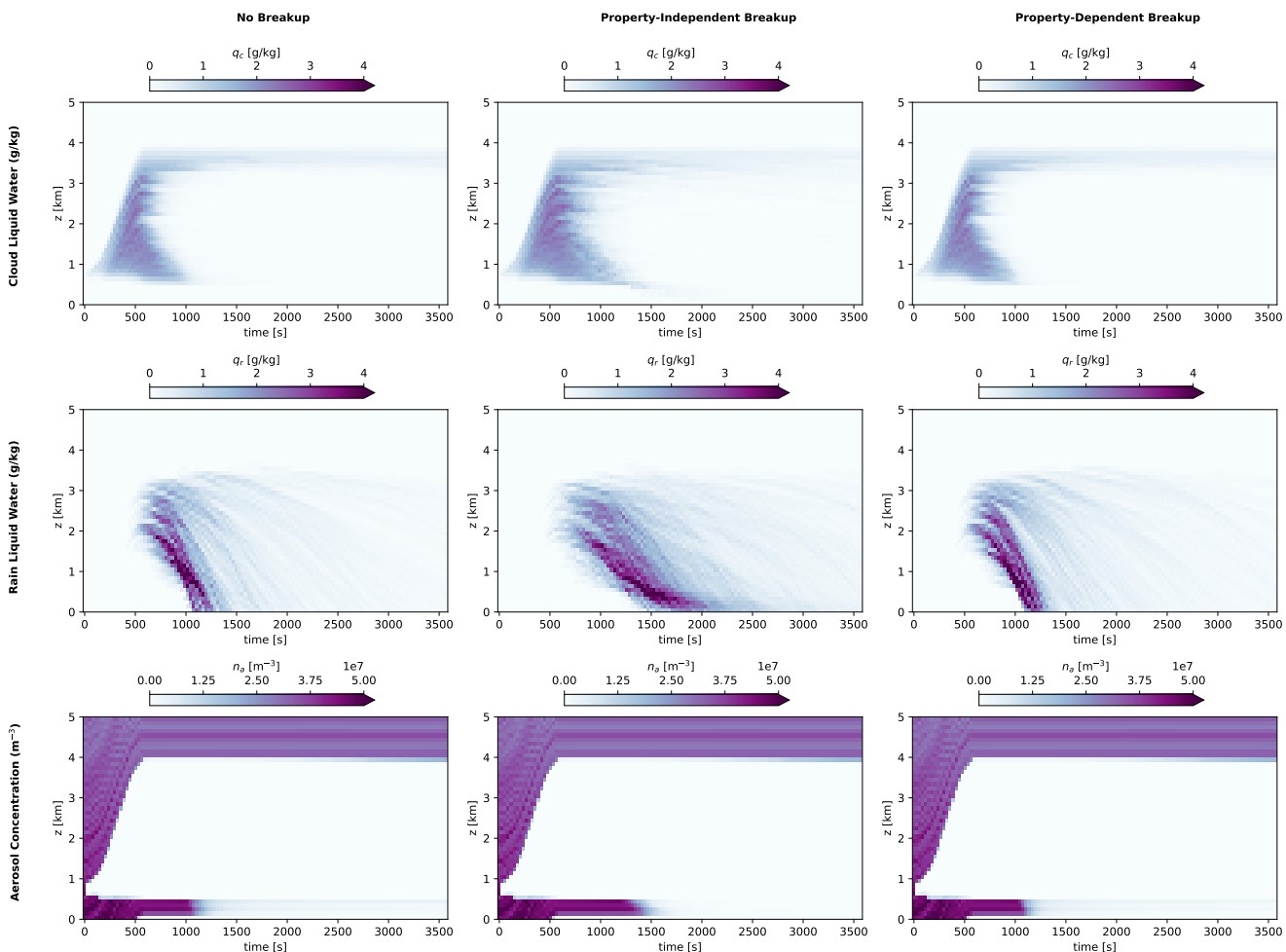

**Figure 10.** Hydrometeor concentrations without breakup (left column), with breakup using a property-independent coalescence efficiency (middle column), and with breakup following the property-dependent Straub et al. (2010) parameterizations (right column). Included are cloud water mixing ratio (top row), rain water mixing ratio (middle row), and aerosol number concentration (bottom row).

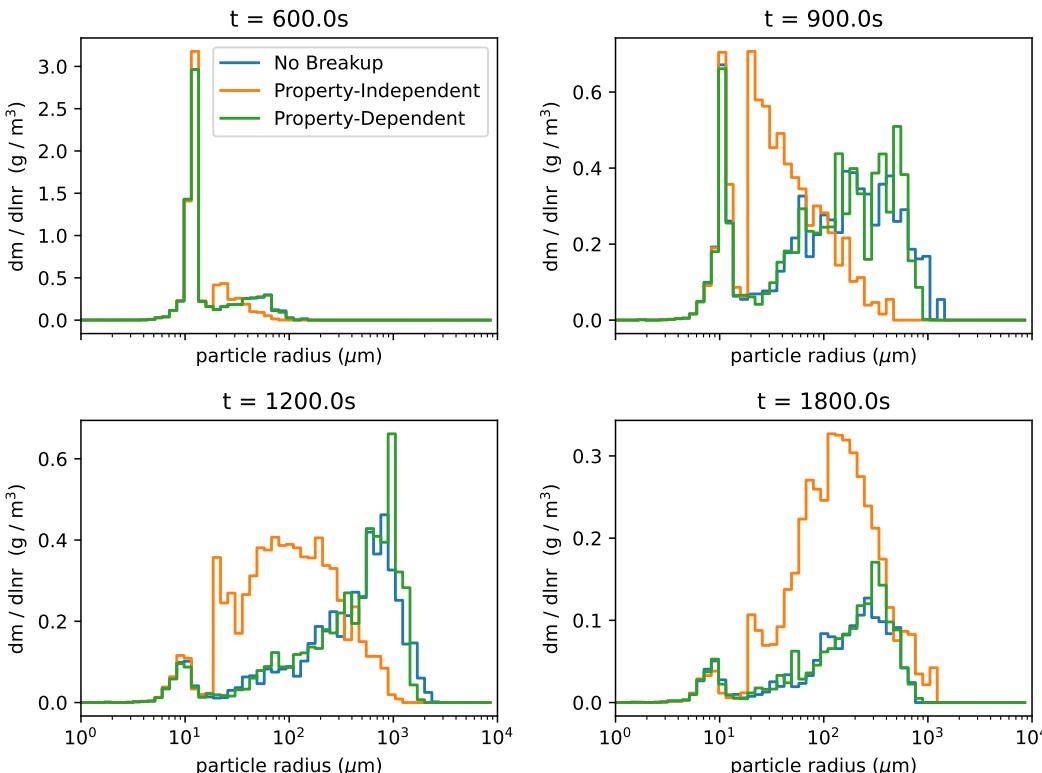

**Figure 11.** Vertically-averaged particle size spectra (as a marginal mass distribution) at four times selected from the 1D rainshaft simulations for each set of dynamics (colors).

overestimates the rate of collisional breakup when condensation and evaporation are present (contrasted with Figure 6). As in the no-breakup case, the property-dependent empirical case displays distinct streaks of precipitation, with surface precipitation initiated around 1100s. While the hydrometeor populations show only slight differences between no-breakup and property-dependent breakup in Figure 10, the size spectra at 900s in Figure 11 shows a somewhat narrower size distribution for the property-dependent case as particles approach the rain size range, suggesting the role of collisional breakup.

The relatively short updraft time and simple one-dimensional representation of this setup produce a short-lived cloud that is precipitating for only a few minutes. The likelihood of breakup in the Straub parameterization is strongly correlated with the size of the colliding droplets, with $E_c$ approaching one for colliding droplets smaller than 1mm diameter, therefore we expect to see a stronger impact of including SDM breakup in a strongly precipitating convective case. Due to the complexity and feedbacks inherent to representing a superdroplet-coupled flow field as well as mixed-phase processes, such experiment is beyond the scope of the present work focused on the algorithm formulation. In deeper mixed-phase clouds, however, secondary ice production via ice-ice and ice-supercooled-liquid collisions are two analogous processes of collisional breakup that may be important (Hallett and Mossop, 1974; Harris-Hobbs and Cooper, 1987; James et al., 2021; Zhao and Liu, 2022). Multiphase

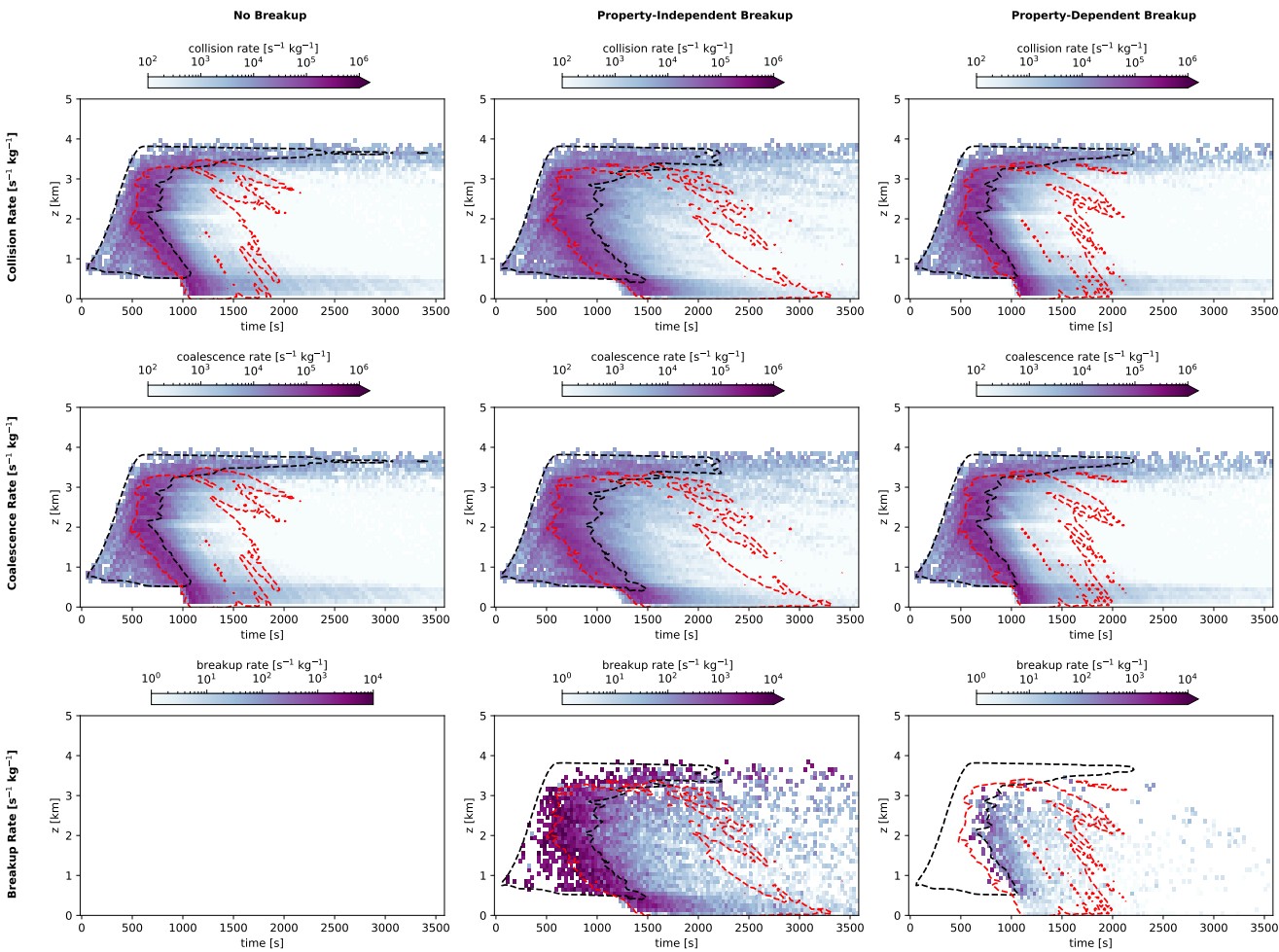

**Figure 12.** Collisional dynamic rates for the 1-dimensional case with (left) no breakup, (middle) breakup with a fixed coalescence efficiency, and (right) breakup with the Straub 2010 parameterizations. The dynamics shown include (top to bottom): collision rate, coalescence rate, and breakup rate. Dashed contour lines represent the level of $q_c = 0.4\mathrm{g/kg}$ (black) and $q_r = 0.4\mathrm{g/kg}$ (red), representing a cloudy and rainy region of the time-space domain, respectively.

superdroplet representation of these mechanisms will face a similar challenge of representing many different fragment sizes, thus the SDM breakup representation presented in this work could be extended to collisional ice processes in future research on secondary ice production and mixed-phase clouds.

**4.2.2 Process Rates**

Figure 12 displays the local rates of superdroplet collision (scaled by multiplicity), as well as distinguishing between rates of coalescence and rates of breakup. We see an expected correlation between the time and location of collisions in all three

cases with the location of hydrometeors (outlined in black for cloud and red for rain)–as expected, a higher concentration of hydrometeors, particularly large rain-range hydrometeors corresponds to higher rates of all collisional dynamics. The rate of

collisions increases throughout the simulation time, particularly near cloud base where the largest droplets are sedimenting and colliding at higher rates. The property-independent case is consistent with the other cases in displaying higher collision rates at cloud base, even though the droplets in this region are slightly smaller and fall in the cloud category.

All three cases display similar rates of collision and coalescence, with the highest of these rates occurring in the cloud among rain droplets, and below cloud base among precipitating rain droplets. In the no-breakup case, every feasible collision results

in a coalescence, and the breakup rate is zero. When property-independent breakup is included, the time-space distribution of the breakup rate is nearly identical to that of the collision rate. This trend is consistent with the use of a uniform coalescence efficiency $E_c = 0.95$, which is agnostic to the size of the colliding particles. In contrast, the empirical property-dependent case sees collisional breakup primarily where larger rain droplets are present, consistent with the Straub et al. parameterization based on Weber number. Breakup events drop off quickly after the initial depletion of the cloud-sized droplets, as the coalescence

efficiency for two large particles is much smaller than that of a small particle colliding with a large particle. As seen in Figure 11, breakup plays less of a role in determining the hydrometeor populations after 1200s of simulation time. Thus, the impacts of collisional breakup are limited in time and space to where large concentration of hydrometeors of both cloud and rain size are colliding. These results demonstrate that the SDM breakup algorithm can produce expected process behavior in both a property-independent setting, where the collision dynamics result in strong breakup, and in an empirically parameterized

setting.

As noted in the discussion of Figure 10, the property-independent breakup case experiences a persistent population of aerosols below cloud base until 1500s, while the no-breakup and property-dependent cases demonstrate washout upon the earlier onset of precipitation. Collisional breakup resulting in very small droplet fragments could potentially introduce cloud droplets so small that they deactivate in their environment. In Figure 13, we investigate whether collisional breakup can induce

significant changes to aerosol processing rates, considering aerosol activation, deactivation, and ripening.

The property-independent and no-breakup cases have nearly identical behavior in aerosol processing, consistent with the correspondence between their hydrometeor concentrations and collision process rates. In all three cases, a few superdroplets at cloud top encounter humidity close to their critical supersaturation, which results in the "ripening" processes of fluctuation between an activated and deactivated state due to competition when the supersaturation is insufficient to activate all aerosols

(e.g., Arenberg, 1939; Wood et al., 2002). (We define ripening rate as the number of activated droplets growing through condensation per unit of time within a grid cell in which deactivation simultaneously occurs on other particles). Aerosols activate primarily at the start of the simulation when an updraft is present, defining the altitude boundaries of the cloud. No additional activation is seen in either instance including collisional breakup. Deactivation occurs among a few aerosols which activate and then rise in altitude beyond cloud top initially, and more strongly below cloud base as droplets sediment out of the cloud

and evaporate. The property-independent case experiences much stronger deactivation at cloud base, which corresponds to the higher rate of fragmentation of droplets at this altitude. The no-breakup and property-dependent breakup cases display continued deactivation of aerosols at the cloud base height throughout the simulation, while the property-independent case shows

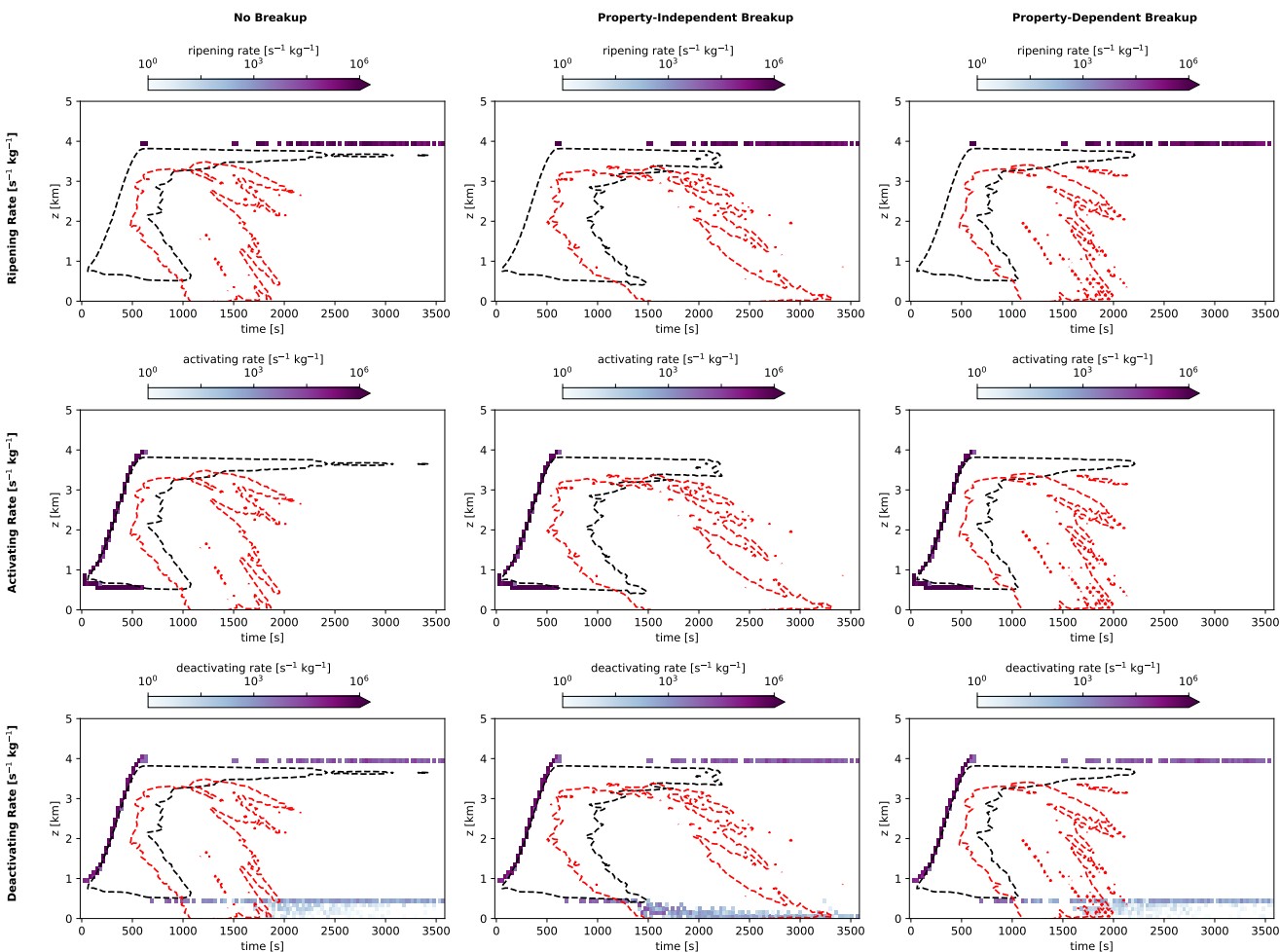

**Figure 13.** Aerosol processing rates for (left to right) no breakup, property-independent breakup, and Straub et al. (2010) parameterizations. Included are (top to bottom): ripening rate, activating rate, and deactivating rate. Dashed contour lines represent the level of $q_c = 0.4\text{g/kg}$ (black) and $q_r = 0.4\text{g/kg}$ (red), representing a cloudy and rainy region of the time-space domain, respectively.

only near-surface deactivation, suggesting aerosol scavenging by rain droplets throughout the more sustained precipitation process. These results indicate that collisional breakup could be a relevant process for future studies of aerosol-cloud effects, particularly in deeper-convective cases where collision rates are likely to be higher.

## 5 Conclusions

This work presents a superdroplet algorithm for collisional breakup that is scalable in avoiding creation of new superdroplets and physical in its ability to produce results in a box and one-dimensional setting that are consistent with the expected reduction and delay in rain formation. Furthermore, the algorithm produces hydrometeor populations and process rates that differ between a property-independent approach (with a fixed coalescence efficiency and fixed fragment size distribution), and a property-dependent approach using empirical parameterizations. These differences indicate the importance of random, stochastic events in warm rain microphysics, a trait which has also been documented in other microphysical phenomena such as giant CCN (Feingold et al., 1999; Yin et al., 2000). Without a scalable representation, the superdroplet method has heretofore been unable to capture these additional stochastic impacts of breakup, nor has it been applied to compare empirical parameterizations of coalescence and breakup, which contribute uncertainties to operational process, weather, and climate models.

This work provides the basis for a more complete representation of microphysical processes in particle-based simulations. For instance, a superdroplet representation that considers ice-phase hydrometeors and properties could probe processes of secondary ice production such as fragmentation and freezing of supercooled water droplets upon collision with ice (James et al., 2021), rime splintering (Hallett and Mossop, 1974), or ice-ice collisions (Harris-Hobbs and Cooper, 1987; Phillips et al., 2017). These and other mixed-phase processes are poorly understood due to challenges in obtaining direct observational or laboratory measurements, thus a high-fidelity particle-based representation such as the superdroplet method provides an ideal means for studying these phenomena. While the collisional breakup representation presented here does not address underlying uncertainties in parameterization of processes such as collision rates and phase change, it provides a path forward for more rigorous and complete studies of cloud microphysics.

*Code and data availability.* Implementation of this breakup algorithm in the SDM is available at https://doi.org/10.5281/zenodo.7851352. The simulations presented in this work (and all necessary input information) are available in the folder 'deJong_Mackay_2022' at https://doi.org/10.5281/zenodo.7851288. The notebooks in this folder reproduce all results and figures presented in this study, with no external datasets required. The scripts run the relevant model configuration in a matter of minutes and plot the resulting output. All results presented in this paper can be reproduced by one of two means: (1) downloading and installing 'PySDM' and 'PySDM-examples' (e.g. using 'pip install'), and running the notebooks locally; (2) accessing the PySDM-examples repository online and running the examples notebooks in the folder 'deJong_Mackay_2022' on Google Colab. These codes, PySDM and PySDM-examples, are continuously under development at https://github.com/atmos-cloud-sim-uj/PySDM and https://github.com/atmos-cloud-sim-uj/PySDM-examples, and are further documented in a software publication (de Jong et al., 2023).

## Appendix A: Limiters

In implementing collisional breakup for superdroplets, we suggest imposing a few limiters to enforce physical constraints and maintain stability of the code. If the user-selected time step for the SDM implementation is too large, collisional breakup may quickly become a runaway process with superdroplet multiplicities increasingly rapidly and unphysically, leading to numerical overflow. As an example, suppose a droplet of multiplicity $10^2$ should undergo 6 collisional breakups ($\gamma_\alpha = 5$) into 5 fragments each time ($N_{f,\alpha} = 5$): then $\gamma_{jk} = 3906$ and its new multiplicity is $15,525 = \mathcal{O}(10^4)$. Successive collisional breakups between droplets whose multiplicities have grown so rapidly would then lead to exponentially booming multiplicities, and could quickly exceed the maximum representable quantity for the computing machine (overflow). One solution is to set a maximum allowable multiplicity for any superdroplet, and to reject any collisional breakups that would produce a superdroplet exceeding this multiplicity.

In addition, the process of collisional breakup is physically constrained such that the resulting superdroplet (the "fragment") volume should not exceed the volume of the coalesced transition state, nor should it drop below a realistic size for a liquid water droplet (molecule scale, for instance). These physical constraints can be imposed by setting a minimum and maximum allowable fragment size resulting from breakup.

The first of these constraints can then be imposed during computation of $\gamma_{jk}$ within the while loop in equation 7:

$$\xi_k^{\mathrm{new}} \leq \xi_{\max} \tag{A1}$$

where $\xi_{\max}$ is a maximum multiplicity set to prevent overflow. The second two constraints are imposed during the sampling of a fragment size:

$$m_{\min} \leq m_{f,\alpha} \leq m_j + m_k \tag{A2}$$

where $m_{\min}$ is a minimum physically allowed fragment size.

## Appendix B: Sampling from empirical fragment size distributions

Sampling a fragment size $m_{f,\alpha}$ requires selecting at random one $m_{f,\alpha}$ according to its weight in the overall fragment size distribution. For fragmentation functions which are composed of several distinct modes, modes must be additionally mass-weighted according to the mode's associated fragment sizes in order to attain the correct expectation value of fragment mass in the resulting sample. A common way of sampling from simple distributions, such as a normal or lognormal, is to invert the CDF (cumulative distribution function). We will demonstrate how to extend this procedure to an empirical fragmentation function comprised of several modes, which lacks a closed form CDF. For instance, the commonly-used fragmentation function of Low and List partitions the fragmentation physics into three categories, corresponding to filament, sheet, and disk breakup. Each category is then comprised of 2–3 modes, corresponding to smaller satellite fragments and larger remnants of the parent droplets. Similarly, Straub et al. distinguish four categories of fragments, with the fragment size distribution within each category following a lognormal or normal distribution. (Here, we follow the notation of Straub et al..)

Suppose the unnormalized fragmentation function $P_f(D)$ in droplet diameter $D$ is described as a sum of $k$ modes:

$$P_f(D) = \sum_{r=1}^{k} N_r p_r(D), \tag{B1}$$

where $N_r$ is the expected number of fragments from mode $r$, and $p_r(D)$ is the normalized fragment size distribution for mode $r$. We begin by re-weighting the size distribution by the mass contained within each mode, $M_r$:

$$M_r = \int_0^{\infty} \frac{\pi}{6} D^3 p_r(D) dD, \tag{B2}$$

and then normalizing the distribution such that its integral is unity:

$$P_{f,n}(D) = \frac{\sum_{r=1}^{k} N_r M_r p_r(D)}{\sum_{r=1}^{k} N_r M_r}. \tag{B3}$$

These initial transformations yield a normalized fragment size distribution which contains all modes of the fragmentation function weighted according to mass. $P_{f,n}(D)$ corresponds to the probability (as a fraction) of retrieving a sample of mass $D$ given the colliding droplet parameters.

Next, to sample a single fragment size $D_f$, we first use the random number $\phi_\alpha''$ to determine which mode of the overall distribution is sampled. This equates to finding $s$ such that

$$\frac{\sum_{r=1}^{s-1} N_r M_r}{\sum_{r=1}^{k} N_r M_r} \leq \phi_\alpha'' < \frac{\sum_{r=1}^{s} N_r M_r}{\sum_{r=1}^{k} N_r M_r}. \tag{B4}$$

The resulting fragment will represent only a single mode of the overall fragment size distribution. However, selecting the mode according to its mass-weighted probability conserves the expected mass distribution of the fragmentation function. The average of several such steps is more likely to sample from each mode, therefore it is crucial that a small enough time step is chosen to allow convergence of this stochastic selection across collisions.

Next, the fragment size is chosen by sampling at random from the CDF of the mode $p_s(D)$, which is assumed to be approximable by a closed form equation (as in the case of a Gaussian or lognormal distribution). This second step of sampling can be accomplished by selecting a new random number, reusing the random number from a different colliding droplet pair, or simply rescaling $\phi_\alpha''$ as

$$\tilde{\phi}_\alpha'' = \frac{\phi_\alpha'' - \sum_{r=1}^{s-1} N_r M_r}{\sum_{r=1}^{s} N_r M_r - \sum_{r=1}^{s-1} N_r M_r}. \tag{B5}$$

The fragment size $D_f$ is then selected such that

$$\tilde{\phi}_\alpha'' = p_s(D_f). \tag{B6}$$

We note that there are several methods of sampling a fragment size from distributions composed of several modes. The presented implementation is used in generating results in this work, and is included as one such example.

*Author contributions.* EdJ led the code development, generation of results, interpretation, and writing. BM contributed to code development and the underlying methodology. OB contributed to algorithm and code review, validation against analytic solution and to porting the code for the GPU backend of PySDM. AJ contributed to underlying methodology and interpretation of results. SA contributed to the code development, interpretation, and leads maintenance of the PySDM codebase.

490   *Competing interests.* The authors declare there to be no competing interests present related to this work.

*Acknowledgements.* We thank Tapio Schneider and Shin-ichiro Shima for feedback, insights, and discussion. The study benefited from peer-review comments provided by Axel Seifert, Christoph Siewert and two anonymous reviewers. Additional thanks go to Piotr Bartman for ongoing support of PySDM. E. de Jong was supported by a Department of Energy Computational Sciences Graduate Fellowship. S. Arabas and O. Bulenok acknowledge support from the Polish National Science Centre (grant no. 2020/39/D/ST10/01220). This research was addi-
495   tionally supported by Eric and Wendy Schmidt (by recommendation of Schmidt Futures) and the Heising-Simons Foundation.

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
