# Peer review of "Breakups are Complicated: An Efficient Representation of Collisional Breakup in the Superdroplet Method"

_EGUsphere, 2022_

## Referee Comment (RC1)

Review: **Breakups are Complicated: An Efficient Representation of Collisional Breakup in the Superdroplet Method**

**Emily De Jong et al.**

The study presents development and preliminary results of drizzle/precipitation collisional breakup (CB) process in the framework of a particle-based ('superdroplet method', SDM) microphysical scheme. This process is physically essential to balance particle growth via collision-coalescence (CC). Including the CB process extends the CC implementation of Shima et al. (2009) in 'PySDM' code base.

The study presents results of an idealized monodispersed drop size distribution evolution under the combined effect of CC and CB, including sensitivity tests of conceptually simpler collisional breakup statistics. The results show that property-dependent CB can eventually balance the CC process which produces a near steady-state size distribution. The CB development was further tested in conjunction of other relevant warm-phase microphysical processes using a 1D rain-shaft setup, where it mainly shows CB is less important in shallow clouds.

The manuscript is reasonably well written and the interaction between various processes is mostly clear.

There is clearly a need for systematic development and documentation of the CB process for more accurate microphysical representation. This marks the importance and motivation of the study, and so the manuscript contributes to the open-source community of the particle-based microphysics. However, the study still shows very preliminary results which are not well justified in relevant cloud setup, and so it marks an incremental step on a path forward.

Based on the above and the specific comments below, I recommend acceptance with Major Revision.

Major comments:

- **Stochastic sampling of the fragment distribution.** Existing fragment size distribution parameterizations of Low and List (1982) and McFarquhar (2004) strongly suggest that each fragment regime (filament, disk, sheet) has underlying physics which is captured by the distinct size-modes in fragments distribution (McFarquhar (2004) section 2a).
  In this case, in Appendix B it's not clear how using a Cumulative Distribution Function as a function of size, rather than the fragments distribution themselves, you intend to capture well these potentially distinct and important modes.

- **Decision pathway, Figure 2 diagram** (L69-L75): To prevent defining a new superdrop size category, you eliminate completely the smaller superdrop (receiver), and so it is now treated as superdrop size category that holds all the 'satellite' unified droplet fragments - is this correct? [Figure 2, lower right arrow, lower/smaller superdrop].

If so, this is different from suggested by Low & List and applied by Seifert et al. (2005) and/or McFarquhar (2004), where the two remanent drops per breakup even are not eliminated.

Thinking about a deeper convective setup with sub-cm/cm -size drops in mind: your algorithm eliminates completely these huge (receiver) drop. These drops are quite low in concentration but should have significant effects over fields like drizzle/precip radar-reflectivity and differential-reflectivity. You should mark this as an assumption to be justified / preliminary results.

- **Abstract / Conclusion** (L310). The term 'rain suppression' is used in the abstract and conclusion (elsewhere) in a way it might be seen as one of the primary goals of the study. First, the term 'rain suppression' is mostly used in Atmospheric science to reflect increase in aerosol loading, followed by increase in cloud droplet number concentration. This has both microphysical (adjustments) and radiative implications.
  Second, CB is an integral physical and mathematical part of the overall CC process, and thus it needs to be seen as an essential complementary process that delays precipitation growth due to CC.

  Both CC and CB clearly depend on physical properties of two interacting drops, hence the importance of the study is in determining realistically what are the relative roles of possibly opposing effects like large/small relative terminal velocity, collision efficiency, coalescence efficiency and characteristic fragments number and size at any such collisional even. The result (outcome) might than show: physically-based delay in growth rate of drizzle/precipitation -size particles a part of the CC process. At the limit of given 'enough' time for CC, the solution converges to near steady-state size distribution. This describes more reliably the presented results.

- **Abstract / Conclusion** (around L320) / **L260 / elsewhere**. The authors proposed the CB algorhytm "to be instrumental in further research on secondary ice production and mixed-phase processes". This is unnecessary and unjustified stretch.
  - First, the proposed CB algorithm/assumptions, being an integral development/part of the CC process, are not validated even for relatively simple warm-phase 1D ('rain-shaft') setup.
  - Second, referring to the Phillips et al. (2018) secondary ice production (SIP) suggested mechanism: the proposed SIP is primarily related to the process of supercooled drops freezing, during which part of the frozen shell fragments to produce ice-splinters (see the diagram in his Figure 7).
    Moreover, since the probability for heterogenous freezing increase with drops volume, the freezing of 'satellite' (small) droplets fragments after collisional breakup are significantly less likely to happen in the relative warmer section of the mixed-phase region, for which the SIP mechanism is suggested
  - Third, the fragmentation discussed in this study results from different underlying physical mechanism compared to the freezing-drop fragmentation process (mode-

1, section 5 in his paper). The fragmentation resulted from collisions between frozen-drops (denser) and more fragile (less dense) ice particles like graupel/ ice/snow (mode-2), resulted primarily from the difference in terminal velocities. Hence, a dedicated microphysical model needs to predict simultaneously these degrees of freedom correctly as a function of modal size and density, which are far more complex than described in this manuscript

- Equation 6 (around L117): It is not clear how multiplicity, being equivalent to number concentration, can be equal to zero. I understand the sink term of the collision-coalescence can (potentially) deplete all the droplets within a superdrop category, where in that case it can be used as a criterion for sub-stepping. But then why you reinitialize the multiplicity with the one from the larger size superdrop category. Please explain.

Minor comments:

L62: The word 'scaling' here refers to computation efficiency. You have used this word for mathematical scaling as well. Please clarify.

L136: Why is that? Is this a choice for computational efficiency, or currently a specific limitation?
This suggest 'PySDM' cannot use collision kernels with turbulent enhancement effects reflecting real clouds, and hence cannot represent potentially important drizzle/precipitation acceleration processes. A specific feature of that acceleration is CC of comparable size drops at the vicinity of turbulent eddies.

Figure 3: The 'units' of ln(R) (natural log of the radius) doesn't have physical units. It is sufficient to mention the spacing is in natural log. Please remove it.

L159: Please remove stochastic

L166: How can one see any manifestation of stochasticity in Figure 3, except for evolution of DSD with two distinct fragmentation distribution? We see different sampling strategies.

Figure 3: The remapping of the superdrop phase space to 128 size bins looks quite wiggly, and probably would need some attention once you compared to observed DSDs.

L177: Droplets with similar size has, by definition, similar terminal velocity. You might change to 'comparable/close in size'.

L219: Please indicate where the microphysical processes algorithms come from (reference/s)?

L255-L260: Please include short discussion on the practical/minimal size range where Straub et al. (2010) can be considered relevant/active.

L258-L259 and elsewhere: It is written in multiple places (including pointing out to various references) that 'Superdrop' / SDM is 'high-fidelity' both in warm-phase and mixed-phase. In that case, except for scalability issues which are less relevant in case of 1-D/'rain-shaft' or 2-D model setups, it's not clear what are the challenges and complexity that prevents one from comparing development work to obs using idealized setup.
This is a minor comment given the manuscript clearly indicates this development work is preliminary incremental path forward subjected to validation.

Figure 7: The separate collision and coalescence panels are redundant, as we saw similar drizzle precip mass in Figure 6. Maybe a different colormap/scale will help. Moreover, for an overlapping single contour of rain and cloud, one cannot relate the rate to specific cloud/rain regime.

Last comment:

I'm relatively new to working with the SDM microphysics, but I have some experience with Seifert et al. (2005) collisional breakup parameterization implemented in a spectral bin microphysical scheme. The figure below depicts a fully-interactive 3D model with basic/medium -complexity mixed-phase microphysics, tested in an idealized 3D squall-line with 120-m/1-km vertical/horizontal resolution (idealize in the sense it simulates a section of a much larger midlatitude squall-line).
Comparing 100 random samples of surface precipitation size distribution from the stratiform area (in both model and obs), the results (yet to be published) shows reasonable realistic comparison.

I would be happy to see and experience comparable setups / results using any SDM code base. Meanwhile, Shima et al. (2020) wrote wise words to manage our expectations and modesty:
"A more detailed evaluation of the model to explore the applicability of the new approach is an essential step forward. Our results strongly indicate that ice particle morphology can be predicted more accurately by further developing particle-based models. However, from this study, we cannot quantify the extent to which the refined representation of mixed phase cloud microphysics could improve the predictability of mixed-phase clouds' macroscopic properties. Such proficiency can be addressed by conducting a thorough comparison with observations and other models"

[Figure]

References:

Bringi, V., Seifert, A., Wu, W., Thurai, M., Huang, G. J., & Siewert, C. (2020): Hurricane Dorian outer rain band observations and 1D particle model simulations: A case study. Atmosphere, 11(8), 879

---

## Referee Comment (RC2)

"Breakups are Complicated: An Efficient Representation of Collisional Breakup in the Superdroplet Method" by Emily De Jong et al.

The article discusses the conceptual picture of collisional breakup and an algorithm for its representation in the Lagrangian particle-based schemes ("superdroplet method"). The proposed numerical implementation of the collisional breakup follows the original algorithm for the collision-coalescence process in the superdroplet method, conserving the number of superdroplets after each event. The fragment sizes are sampled stochastically from idealized/empirical fragment size distributions. The authors also performed idealized box and one-dimensional simulations to understand its performance with artificial size-independent and empirical size-dependent fragment size distributions. After the initial transient phase, approximately steady-state droplet size distributions are achieved due to the balance between coalescence and breakup. Moreover, the one-dimensional simulations show a minor difference with and without collisional breakup, likely due to shallower clouds.

The numerical implementation of the collisional breakup is a critical step for further developing particle-based microphysics schemes. Overall, the topic is well presented, and the manuscript is clearly written. However, as Axel Seifert pointed out in his comments, there is a need to clearly separate the steps between the reference approach closer to physics and its numerical simplifications for practical applicability in realistic cloud conditions. It also requires justification of the simplification through the comparison with reference simulations. I think that's the way model development should proceed in general.

The manuscript is well-suited for Geoscientific Model Development. However, it requires addressing the following specific comments:

- Treating the outcome of a filament breakup event through only two size categories for a colliding superdroplet pair (without introducing a new superdroplet) is a significant simplification and physically inconsistent (at least locally). I think Axel Seifert also made a similar comment. Clearly, there is a need to compare the proposed simplification with a reference run without that simplification. Do we get similar results and similar convergence properties for both approaches? Since the paper's primary focus is to introduce a new breakup algorithm, such a comparison is required. It would be difficult to convince readers of the applicability of the proposed algorithm in more realistic cloud simulations without justifying this simplification in a simpler setup like the current one. Implementing an approach where a new superdroplet is created during breakups would be straightforward in the present box or one-dimensional configuration.

- The collisional breakup introduces an additional element of stochasticity through random sampling of a fragment size distribution. Hence, it's essential to know the convergence properties (with the number of superdroplets) of the mean and variance of drop statistics. No such test is presented in the paper.

- The collisional breakup has almost negligible influences on cloud/rain properties in the one-dimensional test presented here due to a shallow cloud condition. The authors could also test the scheme in an idealized two-dimension deep convection with only warm phase physics. It would help understand the performance of the scheme in more realistic dynamics and the influence of associated feedback.

Minor comments:

- Line 159: "…stochastic stochastic sampling…"

- Figure 4b: Why is a higher $E_c$ value (0.99 vs. 0.95) used here than the deterministic fragmentation function case?

- L287: "The property-independent …" Do you mean a "property-dependent" case here?

- L294: Do you mean a "property-independent" case here?

---

## Community Comment (CC2)

**Collisional breakup with constant super-droplet number**

Axel Seifert and Christoph Siewert

Deutscher Wetterdienst, Offenbach, Germany

January 2023

Here we describe some results of numerical experiments with different implementations of collisional breakup in a super-droplet code of the collision-coalescence-breakup equation. This contributes to a discussion of the manuscript *'Breakups are Complicated: An Efficient Representation of Collisional Breakup in the Superdroplet Method'* (de Jong et al., 2022). The goal is an implementation of collisional breakup with a constant number of super-droplets (SDs). Due to the nature of collisional breakup to produce many droplets of very different sizes from a single collision event, such an implementation is not straightforward.

**Collisional breakup in McSnow**

The Monte-Carlo Lagrangian particle model McSnow of Brdar and Seifert (2018) is based on the super-droplet algorithm of Shima et al. (2009). In McSnow collisional breakup of raindrops is implemented as described in the Appendix of Bringi et al. (2020). The implementation makes use of the empirical parameterizations of Low and List (1982) in the formulation of McFarquhar (2004). Using the equations of McFarquhar (2004) greatly simplifies the implementation compared to the original Low and List scheme. The implementation as described in Bringi et al. (2020) does create new super-droplets during collisional breakup. This seems necessary, especially because it is in fact only one super-droplet available to describe the outcome of the breakup event. Given the fact that breakup can lead to two or three distinct fragment modes, it seems rather difficult to achieve an accurate description with a constant number of SDs. In McSnow a merging algorithm limits the number of SDs afterwards. This requires some additional tuning of the criteria of the merging algorithm. The merging also adds some computational cost which scales with $\mathcal{N} \log(\mathcal{N})$, where $\mathcal{N}$ is the number of SDs in a grid box. Hence, a breakup algorithm with a constant number of SDs would be desirable.

For a given collision of two droplets with drop diameters $D_1$ and $D_2$, masses $m_1$ and $m_2$ and multiplicities $\xi_1$ and $\xi_2$, the super-droplet implementation of collisional breakup provides the masses (and diameters) and the number of the collision fragments in each mode of the fragment distribution function (FDF). For filament breakup these are three fragment modes for the other breakup types (disc and sheet breakup) two different fragment modes occur. The diameters $d_i$ of the collision fragments are sampled from the empirical FDF. The mass of the drop during (temporary) coalescence is $m_{\text{coal}} = m_1 + m_2$ with a corresponding diameter $d_{\text{coal}}$.

**Implementation of constant-SD breakup in McSnow**

Here and in the following we assume $\xi_1 > \xi_2$. Then the SD with diameter $D_1$ is used for the $\xi_1 - \xi_2$ drops that do not participate in the collision event. Hence

$$m_1' = m_1 \tag{1}$$
$$\xi_1' = \xi_1 - \xi_2 \tag{2}$$

where the prime denotes quantities after the collision event and $m$ is the drop mass.

The main idea of the constant-SD breakup is a stochastic sampling of the fragment mode. Here we test two implementations. First, a number-weighted probability sampling as suggested by de Jong et al. (2022). An alternative is a sampling that is proportional to the mass of the fragments. Hence, bigger drops are chosen more often. A third viable implementation would be an equal probability sampling of the fragment modes, but this ends up being quite similar to the number-weighted algorithm and is therefore not included in the following discussion.

The number of fragments in each fragment mode is $n_i$ and the diameter, which has already been sampled from the FDF for that fragment mode, is $d_i$. Note that $n_i$ can be zero if a fragment mode does not exist (sheet and disc breakup). The mass-weighted sampling is implemented as given in Algorithm 1. This stochastic sampling provides the index $i$ of the fragment mode that is then represented by the super-droplet.

For both, number-weighted and mass-weighted sampling, the mass and multiplicity of the second super-droplet is then specified as

$$m'_2 = \frac{\pi}{6} \rho_l \, d_i^3 \tag{3}$$

$$\xi'_2 \; = \frac{m_{\mathrm{coal}}}{m'_2} = \frac{m_1 + m_2}{m'_2} \tag{4}$$

where $\rho_l$ is the density of liquid water. The multiplicity $\xi'_2$ has to be specified by the coalesced mass to ensure mass conservation. Hence, the only information that survives from the breakup parameterization is the diameter $d_i$ that has been sampled from the corresponding mode of the FDF and - more subtle - the probability of occurence of that mode and that specific drop size.
* * *
**Algorithm 1:** Stochastic fragment mode selection in mass-weighted constant-SD breakup
* * *
**1** $p_1 = n_1(d_1/d_{\mathrm{coal}})^3$
**2** $p_2 = n_2(d_2/d_{\mathrm{coal}})^3 + p_1$
**3** call random(r)                 ; uniform random variable in [0,1]
**4** **if** $r < p_1$ **then**
**5**    |   i = 1
**6** **else if** $r < p_2$ **then**
**7**    |   i = 2
**8** **else**
**9**    |   i = 3
* * *
On one hand, the mass-weighted sampling may look somewhat arbitrary at first, but on the other hand the number-weighted sampling can sometimes produce an unreasonable large number of small drops because the coalesced mass $m_{\mathrm{coal}}$ has to be represented by very many tiny droplets of the same size. The fact that $\xi'_2$ is specified by the coalesced mass is another argument to favor the mass-weighted sampling. But *a priori* it is not obvious whether number-weighted or mass-weighted sampling will give the correct solution, if any of the two.

In the following, we compare with the merging-based breakup algorithm of McSnow that does no stochastic mode selection and simply creates super-droplets for all fragments. Those can then be merged in a second step to limit the number of super-droplets. Hence, the merging-based breakup algorithm can serve as a benchmark simulation, especially if the number of super-droplets is large.

**Collisional breakup simulations with McSnow**

We present zero-dimensional box model simulations of the coalescence-breakup equation (similar to section 3.1 of de Jong et al. (2022). The initial condition is a Gamma distribution in drop

mass with a shape parameter of 1, a liquid water content of 2.5 g/m$^3$ and an inital mean mass of $\bar{m}_0 = 3 \times 10^{-10}$ kg. Simulations are performed for 7200 s with a time step of 1 s. The binning of the drop size distribution is done with the kernel density estimator described in Section 5.1.4 of Shima et al. (2009) using $\sigma_0 = 0.62$.

Figure 1 shows the quasi-equilibrium drop distributions reached after 7200 s as number and mass density distributions for different number of super-droplets (SDs) ranging from $\mathcal{N} = 10^6$ over $\mathcal{N} = 4192$ to $\mathcal{N} = 120$. The notation $f(D)$ and $g(\ln r)$ follows Berry and Reinhardt (1974) and $g(\ln r)$ should be equivalent to the $\mathrm{d}m/\mathrm{d}\ln r$ distribution of de Jong et al. (their Figs. 3-5).

The results for $f(D)$ of the merging-based algorithm and the mass-weighted constant-SD algorithm compare reasonably well with each other and with Figure 15 of McFarquhar (2004). In the simulations with high SD number, the mass-weighted constant-SD algorithm exhibits a more pronounced peak from small (filament) droplets compared to the merging-based simulations. Since there is no convergence of the two algorithms to the same solution, it is not clear which simulation is better. The large drop tail of the distributions agrees remarkably well for those two algorithms. For $\mathcal{N} = 120$ both algorithms show a smaller slope of the drop distribution with much larger drops. The number-weighted sampling, on the other hand, gives a much too narrow distribution and does not even converge to a similar distribution as the other two algorithms. This suggests that the number-weighted sampling is generally biased towards small droplets and does not give the correct solution. Hence, the mass-weighted stochastic mode selection seems to be the method of choice for a constant-SD implementation of collisional breakup in a super-droplet algorithm.

**References**

Berry, E. X. and R. L. Reinhardt, 1974: An analysis of cloud drop growth by collection: Part I. Double distributions. *J. Atmos. Sci.*, **31**, 1814–1824.

Brdar, S. and A. Seifert, 2018: McSnow: A Monte-Carlo particle model for riming and aggregation of ice particles in a multidimensional microphysical phase space. *J. Adv. in Modeling Earth Systems*, **10**, 187–206.

Bringi, V., A. Seifert, W. Wu, M. Thurai, G.-J. Huang, and C. Siewert, 2020: Hurricane dorian outer rain band observations and 1d particle model simulations: A case study. *Atmosphere*, **11**, 879.

de Jong, E., J. B. Mackay, A. Jaruga, and S. Arabas, 2022: Breakups are complicated: An efficient representation of collisional breakup in the superdroplet method. *EGUsphere*, **2022**, 1–23, doi:10.5194/egusphere-2022-1243.

Low, T. B. and R. List, 1982: Collision, coalescence and breakup of raindrops, Part II: Parameterization of fragment size distributions. *J. Atmos. Sci.*, **39**, 1607–1618.

McFarquhar, G. M., 2004: A new representation of collision-induced breakup of raindrops and its implications for the shapes of raindrop size distributions. *J. Atmos. Sci.*, **61**, 777 – 794.

Shima, S.-i., K. Kusano, A. Kawano, T. Sugiyama, and S. Kawahara, 2009: The super-droplet method for the numerical simulation of clouds and precipitation: A particle-based and probabilistic microphysics model coupled with a non-hydrostatic model. *Quart. J. Roy. Met. Soc.*, **135**, 1307–1320.

[Figure]

Figure 1: Quasi-equilibrium drops size distributions. Number distributions (left), mass distributions (right), the number of SDs decreases from top to bottom.

---

## Author Response (AR1)

**Response to Reviewers**

April 2023

**Summary**

We thank the reviewers again for proposing changes and additional work that have made this article a stronger and more convincing piece of science. Of note, we now include a detailed section 3 (prepared by a newly added coauthor O. Bulenok) that compares SDM results to a set of analytical collision-coalescence-breakup solutions from Srivastava 1982 and discusses the convergence properties of the SDM breakup algorithm in this context. In addition, we attempt to reproduce figure 10 from Straub 2010 and comment on the ability of the SDM algorithm to reproduce this stationary PSD. The sampling appendix describes the necessity of mass-weighting in a multimodal fragment size distribution, as suggested by Axel Seifert. Among several modifications to the text, we have updated figures 1 and 2 to clarify the role of the receiver droplet and correspondence of the fragment size distribution to resulting droplet sizes.

**Anonymous Referee #1**

Stochastic sampling of the fragment distribution. Existing fragment size distribution parameterizations of Low and List (1982) and McFarquhar (2004) strongly suggest that each fragment regime (filament, disk, sheet) has underlying physics which is captured by the distinct size-modes in fragments distribution (McFarquhar (2004) section 2a). In this case, in Appendix B it's not clear how using a Cumulative Distribution Function as a function of size, rather than the fragments distribution themselves, you intend to capture well these potentially distinct and important modes.

We recognize that our description of how such the fragment sampling step works was misleading. We have made substantial revisions to appendix B (lines 453–454, 469–472) to clarify how the CDF is used to account for all size-modes of the distribution. We concede that there are other means available to sample from the fragmentation functions (lines 480–481).

Decision pathway, Figure 2 diagram (L69-L75): To prevent defining a new superdrop size category, you eliminate completely the smaller superdrop (receiver), and so it is now treated as superdrop size category that holds all the 'satellite' unified droplet fragments - is this correct? [Figure 2, lower right arrow, lower/smaller superdrop]. If so, this is different from suggested by Low & List and applied by Seifert et al. (2005) and/or McFarquhar (2004),

where the two remanent drops per breakup even are not eliminated. Thinking about a deeper convective setup with sub-cm/cm -size drops in mind: your algorithm eliminates completely these huge (receiver) drop. These drops are quite low in concentration but should have significant effects over fields like drizzle/precip radarreflectivity and differential-reflectivity. You should mark this as an assumption to be justified / preliminary results.

The intuition that the "receiver" droplet is eliminated and instead holds droplet fragments is correct. However, the fragments that it represents are not only satellites, but could also take the size of parent 1' or parent 2' (in figure 1). Figures 1 and 2 have been updated to denote how the potential resulting droplet sizes correspond to the fragment size distribution, and the caption of figure 1 and the text (lines 68–73) now clarify that the fragmented receiver could take the size of the satellites or the larger fragmented parents. Indeed, these larger parent (or receiver) drops might have significant effects. We now clarify additionally in appendix B (lines 469–472) that the modes of a fragment size distribution with multiple modes must be mass-weighted such that these important large droplets are not underrepresented.

Abstract / Conclusion (L310). The term 'rain suppression' is used in the abstract and conclusion (elsewhere) in a way it might be seen as one of the primary goals of the study. First, the term 'rain suppression' is mostly used in Atmospheric science to reflect increase in aerosol loading, followed by increase in cloud droplet number concentration. This has both microphysical (adjustments) and radiative implications. Second, CB is an integral physical and mathematical part of the overall CC process, and thus it needs to be seen as an essential complementary process that delays precipitation growth due to CC. Both CC and CB clearly depend on physical properties of two interacting drops, hence the importance of the study is in determining realistically what are the relative roles of possibly opposing effects like large/small relative terminal velocity, collision efficiency, coalescence efficiency and characteristic fragments number and size at any such collisional even. The result (outcome) might than show: physically-based delay in growth rate of drizzle/precipitation -size particles a part of the CC process. At the limit of given 'enough' time for CC, the solution converges to near steady-state size distribution. This describes more reliably the presented results.

Thank you for pointing out misuse of the term "precipitation suppression" in the manuscript. We have edited the abstract (lines 9–10, 12) and the conclusion (lines 398–400) incorporating some of the suggested terminology changes.

Abstract / Conclusion (around L320) / L260 / elsewhere. The authors proposed the CB algorithm "to be instrumental in further research on secondary ice production and mixed phase processes". This is unnecessary and unjustified stretch. First, the proposed CB algorithm/assumptions, being an integral development/part of the CC process, are not validated even for relatively simple warm-phase 1D ('rain-shaft') setup. Second, referring to the Phillips et al. (2018) secondary ice production (SIP) suggested mechanism: the proposed SIP is primarily related to the process of supercooled drops freezing, during which part of the frozen shell fragments to produce ice-splinters (see the diagram in his Figure 7). Moreover, since the probability for heterogenous freezing increase with drops volume, the freezing of 'satellite' (small) droplets fragments after collisional breakup are significantly less likely to happen in the relative warmer section of the mixed-phase region, for which the SIP mechanism is suggested. Third, the fragmentation discussed in this study results from different underlying physical mechanism compared to the freezing-drop fragmentation process (mode-1, section 5 in his paper). The fragmentation resulted from collisions between frozen-drops (denser) and more fragile (less dense) ice particles like graupel/ ice/snow (mode-2), resulted primarily from the difference in terminal velocities. Hence, a dedicated microphysical model needs to predict simultaneously these degrees of freedom correctly as a function of modal size and density, which are far more complex than described in this manuscript

Thank you for pointing out the inconsistency in our reference to Phillips 2018 and the discussed SIP mechanism. We have updated the reference to Phillips 2017, which describes ice-ice collisions, include additional references to SIP publications, and have similarly clarified the text (lines 350–352) and the conclusions (lines 408–415) to specify the role of collisional breakup in an otherwise complex multiphase process. We remove the claim about SIP from the abstract.

Equation 6 (around L117): It is not clear how multiplicity, being equivalent to number concentration, can be equal to zero. I understand the sink term of the collisioncoalescence can (potentially) deplete all the droplets within a superdrop category, where in that case it can be used as a criterion for sub-stepping. But then why you reinitialize the multiplicity with the one from the larger size superdrop category. Please explain.

The sink term in collision coalescence (or collisional breakup) can indeed deplete all the droplets within a superdroplet category. This criterion is NOT used as a criterion for sub-stepping, in fact, but can and does occur. In the original Shima et al 2009 implementation, a superdroplet whose multiplicity becomes zero is removed from the system. Because our implementation seeks to preserve the number of superdroplets in the system, we instead split the remaining superdroplet into two identical superdroplets (re-initializing with half the multiplicity and the same attributes), as described near line 125. We require integer multiplicities, thus if the one remaining superdroplet has a multiplicity of only 1, then the second droplet remains at multiplicity zero and is carried around as an inactive tracer.

Minor comments which warrant response:

- L136: Why is that? Is this a choice for computational efficiency, or currently a specific limitation? This suggest 'PySDM' cannot use collision kernels with turbulent enhancement effects reflecting real clouds, and hence cannot represent potentially important drizzle/precipitation acceleration processes. A specific feature of that acceleration is CC of comparable size drops at the vicinity of turbulent eddies. We do not currently include this implementation in PySDM, and neither does the implementation of Shima et al, but it would indeed be possible to do so for turbulent collision kernels. We clarify in the paper now that neglecting these collisions is specific to the kernel used (lines 198–200).

- Figure 3: The remapping of the superdrop phase space to 128 size bins looks quite wiggly, and probably would need some attention once you compared to observed DSDs. We agree, but believe that the resolution is currently sufficient to communicate the key concepts about sensitivity of the algorithm to coalescence efficiency.

- L219: Please indicate where the microphysical processes algorithms come from (reference/s)? We now specify (lines 301-302) that the kappa-Kohler theory is used for aerosol activation and that the procedures are based on those of the program libcloudph++ (10.5194/gmd-8-1677-2015). However, the SDM solves for condensation directly without further simplification or parameterization, thus we have no further references to cite. The implementation of condensation is available in the source code and documentation of 'PySDM' for interested parties.

- L258-L259 and elsewhere: It is written in multiple places (including pointing out to various references) that 'Superdrop' / SDM is 'high-fidelity' both in warm-phase and mixed-phase. In that case, except for scalability issues which are less relevant in case of 1-D/'rain-shaft' or 2-D model setups, it's not clear what are the challenges and complexity that prevents one from comparing development work to obs using idealized setup. This is a minor comment given the manuscript clearly indicates this development work is preliminary incremental path forward subjected to validation. Thank you, we have now clarified lines 348–350. The core challenge is in representing the dynamics of the flow field and coupling evolution of the flow-field to particles, rather than in representing the particles themselves.

- Figure 7: The separate collision and coalescence panels are redundant, as we saw similar drizzle precip mass in Figure 6. Maybe a different colormap/scale will help. Moreover, for an overlapping single contour of rain and cloud, one cannot relate the rate to specific cloud/rain regime. We have decided to include both panels for completeness (now figure 12). The contours have been adjusted as well, and are meant to provide intuition about the presence of hydrometeors rather than characterize specific regimes.

- I'm relatively new to working with the SDM microphysics, but I have some experience with Seifert et al. (2005) collisional breakup parameterization implemented in a spectral bin microphysical scheme. The figure below depicts a fully-interactive 3D model with basic/medium -complexity mixed-phase microphysics, tested in an idealized 3D squall-line with 120-m/1-km vertical/horizontal resolution (idealize in the sense it simulates a section of a much larger midlatitude squall-line). Comparing 100 random samples of surface precipitation size distribution from the stratiform area (in both model and obs), the results (yet to be published) shows reasonable realistic comparison. I would be happy to see and experience comparable setups / results using any SDM code base. Thank you for sharing these interesting results! Unfortunately 'PySDM' does not include coupling to a large eddy simulation or other flow solver at this time, so we would be unable to reproduce your simulation in anything other than a prescribed-flow setting. SDM implementations coupled to flow solvers do exist however, such as SCALE-SDM in Professor Shima's group.

**Anonymous Referee #2**

Treating the outcome of a filament breakup event through only two size categories for a colliding superdroplet pair (without introducing a new superdroplet) is a significant simplification and physically inconsistent (at least locally). I think Axel Seifert also made a similar comment. Clearly, there is a need to compare the proposed simplification with a reference run without that simplification. Do we get similar results and similar convergence properties for both approaches? Since the paper's primary focus is to introduce a new breakup algorithm, such a comparison is required. It would be difficult to convince readers of the applicability of the proposed algorithm in more realistic cloud simulations without justifying this simplification in a simpler setup like the current one. Implementing an approach where a new superdroplet is created during breakups would be straightforward in the present box or one-dimensional configuration.

We now include two comparisons to published data in order to justify and validate the algorithm. The first is presented in a new section 3, and includes comparisons to analytical solutions from Srivastava 1982. The second is included in figure 9, discussed in lines 272–317, and compares the steady state size distribution using the Straub 2010 parameterizations to the results published in the same article. As a rigorous validation of superdroplet-creating representation has not been carried out to our knowledge, we found these comparisons against published results to be the most convenient and convincing. Unfortunately it is not straightforward to implement creation of new superdroplets in the code-base PySDM used for these studies due to its parallel-computing properties.

The collisional breakup introduces an additional element of stochasticity through random sampling of a fragment size distribution. Hence, it's essential to know the convergence properties (with the number of superdroplets) of the mean and variance of drop statistics. No such test is presented in the paper.

Our new comparisons to published data (Srivastava 1982 in section 3, Straub 2010 in figure 9) now plot the mean and spread of the SDM model results using different numbers of superdroplets. A detailed discussion of the convergence properties is likewise presented in section 3.

The collisional breakup has almost negligible influences on cloud/rain properties in the one-dimensional test presented here due to a shallow cloud condition. The authors could also test the scheme in an idealized two-dimension deep convection with only warm phase physics. It would help understand the performance of the scheme in more realistic dynamics and the influence of associated feedback.

We hoped to follow the reviewer's suggestion of an idealized two-dimensional deep convection setting, but were unable to find or replicate a validated prescribed flow setting from the literature. Instead, to address these concerns, we have increased the updraft velocity and domain size of the presented one-dimensional setting to 6 m/s and 5km (respectively). We also now present vertically-averaged particle size spectra sampled at various times during the simulation in figure 10. We discuss in lines 342-344 the implications of property-dependent breakup on the spectra during the brief 5 minute window from 900s to 1200s.

Minor comments warranting response:

- Figure 4b: Why is a higher Ec value (0.99 vs. 0.95) used here than the deterministic fragmentation function case? Thank you for the catch – both cases now use Ec=0.95, and we have switched to an exponential fragment size distribution in 7b (only one parameter instead of two).

---

## Referee Report (RR1)

Review: **Breakups are Complicated: An Efficient Representation of
Collisional Breakup in the Superdroplet Method**

**Emily De Jong et al.**

This is a revised submission. The authors have made efforts within the scope of the study to enhance the coherence and provide more concise explanations regarding the collisional breakup process compared to the previous manuscript.

Considering the minor comments provided below, I recommend acceptance with minor revisions.

Minor comments:

**Introduction:**

- o L37: In principle, it is clear the conservation of total droplet/drop number is fundamentally independent of scalability. These are distinct physical and technical aspects, respectively. It is an optimization problem: (a) the fundamental collisional breakup has the potential to generate new droplet sizes explosively; (b) the physical process needs to conserve mass and number reasonably well; (c) applying (a) & (b) together in an atmospheric numerical model requires some technical adaptation. The manuscript presents an algorithm with some associated assumptions that optimize the above requirements. Please refine/clarify the text.
- o L45: I think that these sentences are written primarily for warm rain scenarios. In modeling mixed-phase microphysics, it becomes evident that large snow aggregates and/or graupel/hail entering the melting layer can lead to the formation of aerodynamically unstable large drops.
  Additionally, due to the short time scale and very (very) low concentration of these drops, applying standard (unit volume wise) collision algorithms may not be fully applicable. These drops are also important for radar-based analysis and are associated with constraining maximal dimension (please refer to the references below). Please refine your text. You may consider simply saying that spontaneous breakup is not included in this version of SDM (as I think you points out later on in the manuscript).
- o L170: Please correct the typo "The simulations are performed for 2048**s with** 1s timesteps …"

**Section 4.1.1:**

- o L240: relative **terminal** velocity

**Section 4.1.2:**

- o L277: Droplets are referred to cloud droplets. 1-mm particles are rain **drops** (throughout the text).
- o **Figure 9**: Overall, I do not see any good reason to keep this figure. Presenting number size distribution clearly needs to have better agreement with the reference's small diameter modal size, particularly in logarithmic scale.
  The significant increase in the number concentration at modal size of ~6-mm is concerning and is physically unrealistic as drops becomes increasingly unstable at these sizes. If the authors were to present the data in terms of mass size distribution, it would demonstrate an explosive mass of (unrealistic) drops. Lastly, the reader lacks an effective means to evaluate why the multiplicity-limiter favors coalescence rather than breakup, which is related to the disclaimer at L289. Out of respect to the authors work, I kindly defer the decision on the relevance of this figure to the authors and/or editor.

**References:**

Kacan, K.G. and Lebo, Z.J., 2019. Microphysical and dynamical effects of mixed-phase hydrometeors in convective storms using a bin microphysics model: Melting. Monthly Weather Review, 147(12), pp.4437-4460; https://doi.org/10.1175/MWR-D-18-0032.1

Carey, L.D. and Petersen, W.A., 2015. Sensitivity of C-band polarimetric radar–based drop size estimates to maximum diameter. Journal of Applied Meteorology and Climatology, 54(6), pp.1352-1371; doi: 10.1175/JAMC-D-14-0079.1

---

## Author Response (AR2)

**Response to Suggested Minor Revisions**

June 2023

**Anonymous Referee #2**

L37: In principle, it is clear the conservation of total droplet/drop number is fundamentally independent of scalability. These are distinct physical and technical aspects, respectively. It is an optimization problem: (a) the fundamental collisional breakup has the potential to generate new droplet sizes explosively; (b) the physical process needs to conserve mass and number reasonably well; (c) applying (a) (b) together in an atmospheric numerical model requires some technical adaptation. The manuscript presents an algorithm with some associated assumptions that optimize the above requirements. Please refine/clarify the text.

Thank you for suggesting this clarification, which we now have incorporated into lines 34–40.

L45: I think that these sentences are written primarily for warm rain scenarios. In modeling mixed-phase microphysics, it becomes evident that large snow aggregates and/or graupel/hail entering the melting layer can lead to the formation of aerodynamically unstable large drops. Additionally, due to the short time scale and very (very) low concentration of these drops, applying standard (unit volume wise) collision algorithms may not be fully applicable. These drops are also important for radar-based analysis and are associated with constraining maximal dimension (please refer to the references below). Please refine your text. You may consider simply saying that spontaneous breakup is not included in this version of SDM (as I think you points out later on in the manuscript).

We now clarify in lines 46–48 that we refer to liquid microphysics and do not include spontaneous breakup.

L170: Please correct the typo "The simulations are performed for 2048s with 1s timesteps ..."

L240: relative "terminal" velocity

Thank you, the corrections have been made.

L277: Droplets are referred to cloud droplets. 1-mm particles are rain drops (throughout the text).

We have chosen to use the term "droplet" throughout the text (and have removed two instances of the term "drop" referring to a hydrometeor) for consistency. Where comments regarding precipitation or a rain size threshold are pertinent (ex. section 4.2.1), we use the

term "rain droplet". Otherwise we prefer to refer to all liquid hydrometeors as droplets considering ongoing debate about size cutoffs in determining hydrometeor categories (see, e.g. Igel et al. 2022).

Figure 9: Overall, I do not see any good reason to keep this figure. Presenting number size distribution clearly needs to have better agreement with the reference's small diameter modal size, particularly in logarithmic scale. The significant increase in the number concentration at modal size of 6-mm is concerning and is physically unrealistic as drops becomes increasingly unstable at these sizes. If the authors were to present the data in terms of mass size distribution, it would demonstrate an explosive mass of (unrealistic) drops. Lastly, the reader lacks an effective means to evaluate why the multiplicity-limiter favors coalescence rather than breakup, which is related to the disclaimer at L289. Out of respect to the authors work, I kindly defer the decision on the relevance of this figure to the authors and/or editor.

Thank you for pointing out these concerns. We have chosen to include figure 9 in the final publication, as we believe it best addresses previous reviewer comments requesting a comparison of our method against a "ground truth". We clarify in line 277 that number distribution is plotted to be consistent with previous publications presenting steady state size distributions. We additionally temper the language about the multiplicity limiter (line 287) and clarify the relevance of the rainshaft simulation results in lines 291-294.

---

## Author Response (AR3)

Please note that the affiliation of two authors (Ben Mackay and Sylwester Arabas) has been updated to reflect their current affiliation, with reference to their affiliation at the time of research. This is the only change that has been made to the manuscript.